# Bone morphogenetic protein 7 promotes resistance to immunotherapy

Maria Angelica Cortez [1✉], Fatemeh Masrorpour[1], Cristina Ivan [2], Jie Zhang[3], Ahmed Younes [1], Yue Lu[4], Marcos R Estecio[4], Hampartsoum B. Barsoumian[1], Hari Menon[1], Mauricio da Silva Caetano[3], Rishab Ramapriyan [1], Jonathan E. Schoenhals[3], Xiaohong Wang[3], Ferdinandos Skoulidis[5], Mark D. Wasley[1], George Calin [2], Patrick Hwu[6] & James W. Welsh[1]

Immunotherapies revolutionized cancer treatment by harnessing the immune system to target cancer cells. However, most patients are resistant to immunotherapies and the mechanisms underlying this resistant is still poorly understood. Here, we report that over-expression of BMP7, a member of the TGFB superfamily, represents a mechanism for resistance to anti-PD1 therapy in preclinical models and in patients with disease progression while on immunotherapies. BMP7 secreted by tumor cells acts on macrophages and CD4[+] T cells in the tumor microenvironment, inhibiting MAPK14 expression and impairing pro-inflammatory responses. Knockdown of BMP7 or its neutralization via follistatin in combination with anti-PD1 re-sensitizes resistant tumors to immunotherapies. Thus, we identify the BMP7 signaling pathway as a potential immunotherapeutic target in cancer.

[1] Departments of Radiation Oncology, The University of Texas MD Anderson Cancer Center, Houston, TX, USA. [2] Experimental Therapeutics, The University of Texas MD Anderson Cancer Center, Houston, TX, USA. [3] Experimental Radiation Oncology, The University of Texas MD Anderson Cancer Center, Houston, TX, USA. [4] Epigenetic and Molecular Carcinogenesis, The University of Texas MD Anderson Cancer Center, Houston, TX, USA. [5] Thoracic/Head and Neck Medical Oncology, The University of Texas MD Anderson Cancer Center, Houston, TX, USA. [6] Melanoma Medical Oncology, The University of Texas MD Anderson Cancer Center, Houston, TX, USA. ✉email: MACortez@mdanderson.org

Although antibodies blocking PD1/PDL1 have led to impressive clinical responses in some patients with melanoma, lung cancer, or renal cell carcinoma, the objective response rates to single-agent anti-PD1 or -PDL1 therapies are only 15–25% in chemotherapy-refractory non-small cell lung cancer (NSCLC)[1,2]. That many patients either do not respond to or develop recurrence after immunotherapy indicates the presence of intrinsic or acquired resistance[3]. This observation raises fundamental questions about mechanisms underlying non-response and potential strategies to overcome anti-PD1/PDL1 resistance—a major unmet therapeutic need. To answer these questions, we previously generated an anti-PD1-resistant preclinical tumor model involving an anti-PD1-resistant variant of the murine lung cancer cell line 344SQ in syngeneic mice[4]. On that study, we found that 344SQR-resistant tumors are enriched in myeloid-derived suppressor cells (MDSCs) with decreased infiltration of CD4+ and CD8+ T cells, and reduced interferon-gamma (IFNG) production[4]. We sought to identify mechanisms of resistance to anti-PD1 therapy and found several promising pathways that have potential as both biomarkers and therapeutic targets to overcome immune evasion. Specifically, we found that anti-PD1-resistant tumors overexpress bone morphogenetic protein (BMP)7, also known as OP-1. BMP7 is a secreted protein that belongs to the transforming growth factor beta (TGFB) superfamily and regulates proliferation, differentiation, and apoptosis in many different cell types by altering target gene transcription. BMP ligands bind to and form heteromeric complexes with two types of serine/threonine kinase receptors on the cell surface, which then activate the "small mothers against decapentaplegic" (SMADs) proteins in cells[5–7].

BMPs can act as either tumor suppressors or oncogenes depending on the cellular context and tumor type. BMP7 has been reported in a wide range of human cancers and is associated with metastasis and poor prognosis[8–13]. In lung cancer, BMP7 overexpression was associated with lymph node involvement and an indicator of bone metastasis[14,15]. The immune-regulatory functions of BMPs are not well understood. Nonetheless, accumulating evidence indicates that BMPs regulate immune cell responses and are immunosuppressive in cancer[16]. For example, BMPs have been shown to regulate activation, growth, and cytokine secretion in macrophages and to promote PDL1 and PDL2 upregulation in dendritic cells[17–19]. Treatment with BMP7 in vitro and in vivo significantly enhanced monocyte polarization into M2 macrophages[20–22]. Here, we report that BMP7 is overexpressed in an anti-PD1-resistant mouse model and in patients with disease that progressed while on immunotherapy. Our findings show that BMP7 regulates proinflammatory responses in the tumor microenvironment by suppressing mitogen-activated protein kinase 14 (MAPK14) signaling in macrophages and CD4+ T cells. Furthermore, BMP7 inhibition in combination with anti-PD1 therapy activates CD4+ and CD8+ T cells in tumors, decreases M2 macrophages, and re-sensitizes resistant tumors to immunotherapies.

## Results

**BMP7 is upregulated in tumors that did not respond to anti-PD1.** We previously generated a syngeneic preclinical model of NSCLC (p53$^{R172H\Delta g/+}$Kras$^{LA1/+}$) with acquired resistance to anti-PD1[4]. In the current study, we investigated methylation differences in specific genomic regions comparing anti-PD1-resistant tumors (344SQR) with their parental-tumor counterparts (344SQP). Overall, genes were hypomethylated in anti-PD1-resistant tumors compared with parental tumors (Fig. 1a). Although some genes such as *KCNK4*, *RAVER2*, and *NAV1* were hypermethylated, others including *BMP7*, *SNORD37*, and

*SLC2a13* were hypomethylated in 344SQR tumors compared with parental tumors (Fig. 1b; Supplementary Data 1). Our initial finding via microarray analysis that BMP7 was one of the top genes upregulated in the anti-PD1-resistant model (Supplementary Table 1) led us to focus here on validating BMP7 as a target for resistance to anti-PD1. We confirmed that the BMP7 promoter CpG is hypomethylated, with an average of 4.28% in 344SQR tumors versus 28.68% in 344SQP tumors (Fig. 1c; Supplementary Table 2). We next validated BMP7 upregulation at the mRNA level in 344SQR and 344SQP tumors using quantitative polymerase chain reaction (PCR) (Fig. 1d). Because BMP7 is a secreted protein, we evaluated BMP7 levels in plasma from mice bearing resistant or parental tumors. We found that BMP7 levels were higher in serum from mice bearing 344SQR tumors than in mice with parental tumors treated with anti-PD1 therapy (Fig. 1e). We then analyzed BMP7 levels in pretreatment plasma samples from patients with subsequent progressive disease (PD) on pembrolizumab versus patients with progressive response (PR) or stable disease (SD) (NCT02444741; NCT02402920). Patients with progression on pembrolizumab had significantly higher levels of plasma BMP7 before treatment than did patients with PR or SD (Fig. 1f). We validated BMP7 upregulation at the protein level in 344SQR and 344SQP tumors (Fig. 1g) and in samples from patients with NSCLC, mixed Mullerian carcinoma, and adrenocortical carcinoma that initially responded to and later progressed on pembrolizumab (NCT02444741) or ipilimumab (NCT02239900) using immunohistochemical staining. We found that BMP7 was overexpressed in the progressed sample (PD) compared with primary tumor (pretreatment) (Fig. 1h–j). Collectively, our findings show that tumors resistant to anti-PD1 had upregulated BMP7 expression and secretion and suggest that its overexpression may promote resistance to immunotherapies.

**BMP7 modulates MAPK14 in tumors and immune cells.** Next, to identify the molecular mechanism by which BMP7 promotes resistance to anti-PD1, we analyzed the expression levels and activation status of 243 proteins in 344SQP and 344SQR tumors treated with anti-PD1. Proteins known to be modulated by BMP7 were found to be expressed at different levels in resistant tumors than in parental tumors. For example, MAPK14 was downregulated and CTNNB1 (β-catenin) was upregulated in 344SQR versus 344SQP. Other downregulated proteins included CDKN2A (p16), PTEN, and granzyme B (GZMB), and other upregulated proteins included HIST3H3, SOX2, and PARP1 (Fig. 2a; Supplementary Table 3). Next, we evaluated if MAPK14 downregulation depended on BMP7 expression. We found that MAPK14 mRNA levels were upregulated in BMP7-knockdown tumors treated with anti-PD1 relative to control tumors (Supplementary Figs. 1 and 11). Because MAPK14 is inhibited by BMP7[23,24] via SMAD1 at high BMP7 concentrations[25], we analyzed MAPK14, SMAD1, and p-SMAD1/5/9 expression in 344SQP versus 344SQR tumors treated with IgG or anti-PD1. We found that 344SQR tumors treated with IgG or anti-PD1 expressed less MAPK14 and had higher activation of SMAD1 than did parental tumors (Fig. 2b, d; Supplementary Fig. 2a). We then evaluated MAPK14, SMAD1, and SMAD1/5/9 activation status in BMP7-knockdown tumors treated with IgG or anti-PD1 and control tumors, and confirmed that BMP7-knockdown tumors had higher MAPK14 expression and lower SMAD1 activation than control tumors treated with IgG or anti-PD1 (Fig. 2c, e Supplementary Fig. 2b). Notably, we next validated BMP7 and MAPK14 expression and SMAD1 activation in samples from patients with NSCLC and adrenocortical carcinoma that progressed in the lung after treatment with pembrolizumab and ipilimumab, respectively. Patients with disease progression

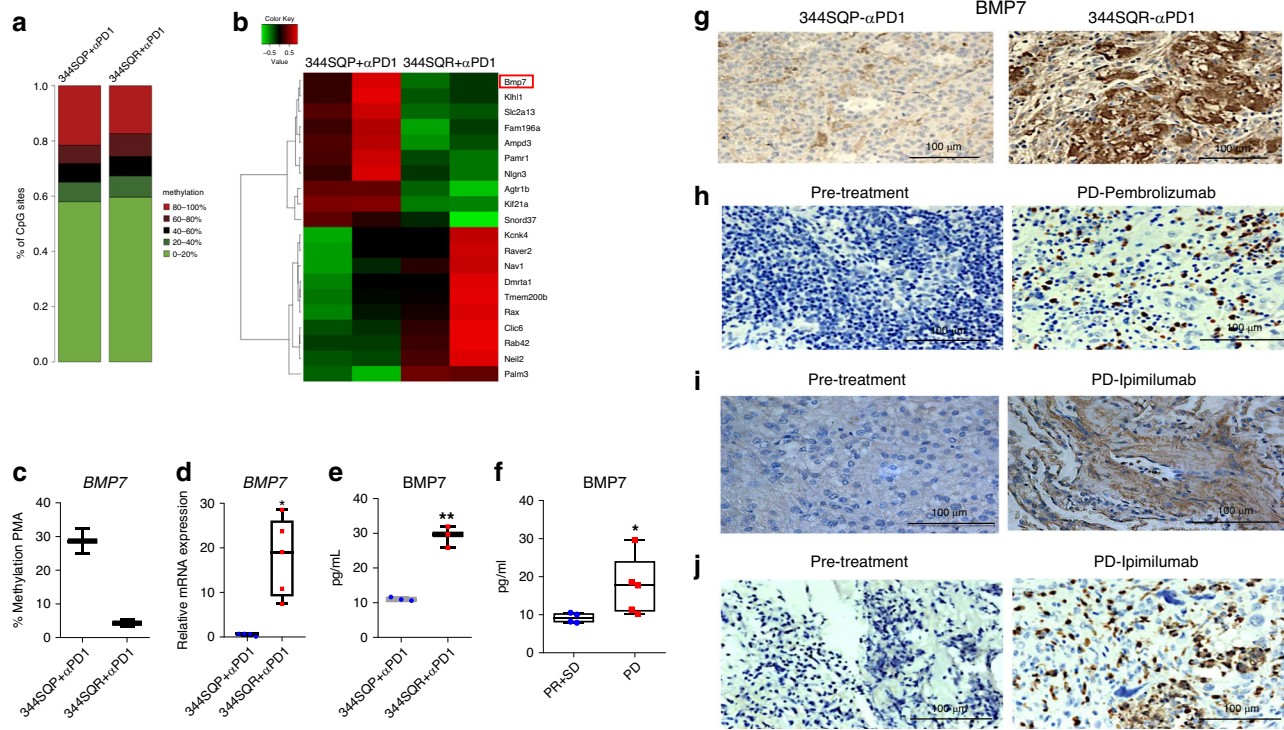

**Fig. 1 BMP7 is upregulated in tumors resistant to immunotherapies. a, b** Reduced representation bisulfite sequencing (RRBS) results from 344SQP (parental) (n = 2 biologically independent samples) and 344SQR (anti-PD1-resistant) (n = 2 biologically independent samples) tumors treated with anti-PD1 (10 mg/kg). **a** Percentages of CpG sites methylation and **b** heatmap of top 10 hypomethylated (green) and 10 hypermethylated (red) genes in 344SQP (n = 2 biologically independent samples) and 344SQR (n = 2 biologically independent samples) tumors treated with anti-PD1(10 mg/kg). The methylation percentages for CpG sites were calculated by the bismark_methylation_extractor script from Bismark and an in-house Perl script. the differential methylation on CpG sites was statistically assessed by R/Bioconductor package methylKit (version 0.9.5). The CpG sites with read coverage ≥20 in all the samples were qualified for the test. The significance of differential methylation on gene level was calculated using Stouffer's z score method by combining all the qualified CpG sites inside each gene's promoter region (defined as −1000bp to +500 of TSS), and was corrected to FDR by Benjamini & Hochberg (BH) method. **c** Pyrosequencing methylation assay with specific primers for BMP7 CpG in 344SQP (n = 2 biologically independent samples with three technical replicates for each sample) and 344SQR (n = 2 biologically independent samples with three technical replicates for each sample) tumors treated with anti-PD1(10 mg/ kg). Box-and-whisker plots show the minimum and maximum values. **d** Quantitative polymerase chain reaction (PCR) analysis of BMP7 expression in 344SQP (n = 4 biologically independent samples with two technical replicates for each sample) and 344SQR (n = 5 biologically independent samples with two technical replicates for each sample) tumors treated with anti-PD1 (10 mg/kg). ACTB expression was used as a housekeeping gene for quantitative PCR analysis. The comparative Ct method was used to calculate the relative abundance of mRNAs compared with ACTB expression. Box-and-whisker plots show the minimum and maximum. **p = 0.0159, two-sided Mann–Whitney test. **e** Enzyme-linked immunosorbent assay of BMP7 levels in serum from mice bearing 344SQR (n = 3 biologically independent samples with two technical replicates for each sample) or 344SQP (n = 3 biologically independent samples with two technical replicates for each sample) tumors treated with anti-PD1(10 mg/kg). Box-and-whisker plots show the minimum and maximum. **p = 0.0080 unpaired, two-sided t tests. **f** Enzyme-linked immunosorbent assay for BMP7 in pretreatment plasma collected from patients with disease that progressed (PD) (n = 5 biologically independent samples) on pembrolizumab (NCT02444741; NCT02402920) versus patients with progressive response (PR) or stable disease (SD) (n = 4 biologically independent samples). Box-and-whisker plots show the minimum and maximum. *p = 0.0317, two-sided Mann–Whitney test. **g** Representative images of immunohistochemical stains of BMP7 expression in formalin-fixed paraffin-embedded tissue sections from 344SQP and 344SQR tumors treated with anti-PD1. Data shown are representative of two reproducible independent experiments. Scale bar, 100 μm (×40 magnification). **h–j** Representative images of immunohistochemical stains of BMP7 expression in formalin-fixed paraffin-embedded tissue sections collected from patients with NSCLC, adrenocortical carcinoma, and mixed Mullerian carcinoma before and at the time of disease progression on pembrolizumab or ipimilumab. Data shown are representative of two reproducible independent experiments. Scale bar, 100 μm (×40magnification).

on immunotherapy expressed higher levels of BMP7 as previously shown in Fig. 1 and activation of SMAD1/5/9 and lower levels of MAPK14 in tumors after immunotherapy than in pretreatment tumors (Fig. 2f, Supplementary Fig. 2c). These findings suggest that BMP7 downregulates MAPK14 via activation of SMAD1 pathway in tumors resistant to anti-PD1 therapy.

Next, we hypothesized that secreted BMP7 negatively affects immune cells in the tumor microenvironment of anti-PD1-resistant tumors. We then analyzed tumor-infiltrating leukocytes (TILs) collected from 344SQP and 344SQR tumors treated with anti-PD1 with the Nanostring Immune Panel. Strikingly, we found that *MAPK14* was also downregulated in TILs from 344SQR tumors relative to parental tumors (Fig. 2g).

Interestingly, we also found different expression levels of *SLC7A11*, *CD274* (PDL1), *NLRP3*, and *MUC1* in TILs isolated from 344SQR versus parental tumors treated with anti-PD1 (Fig. 2g; Supplementary Table 4). We further found that several inflammatory cytokines and genes regulated by MAPK14 were downregulated in TILs from the 344SQR tumors relative to parental tumors, including *IL1A*, *IL1B*, *TNF*, and *ATF1* (Fig. 2g). To validate these findings, we analyzed serum levels of MAPK14-regulated cytokines and chemokines from mice bearing 344SQR or 344SQP tumors. In agreement with our Nanostring data, levels of IL1A, IL1B, and TNF were downregulated in serum from mice bearing 344SQR tumors versus parental tumors (Supplementary Fig. 3; Supplementary Methods). We further found that CCL5

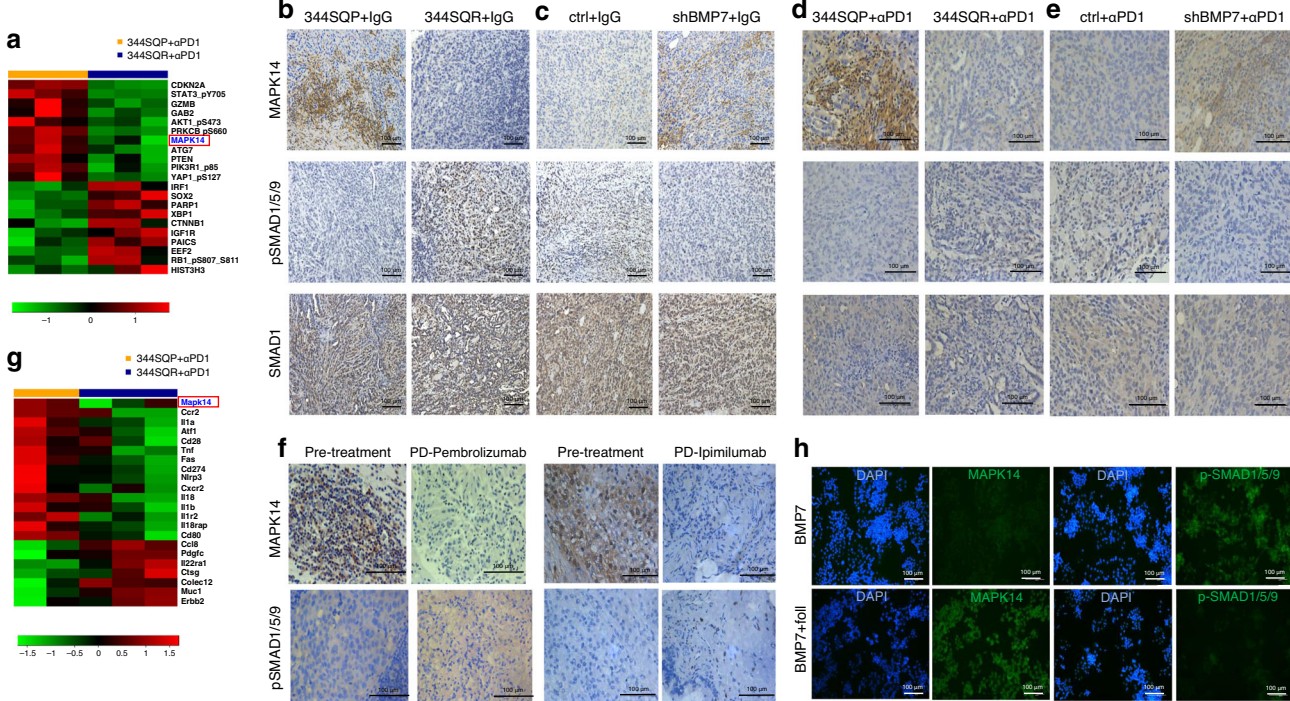

**Fig. 2 BMP7 modulates MAPK14 in anti-PD1-resistant tumors and immune cells. a** Reverse phase protein array (RPPA) results on expression levels and activation status of 243 proteins in 344SQP (*n* = 3 biologically independent samples) and 344SQR (*n* = 3 biologically independent samples) tumors treated with anti-PD1(10 mg/kg). Normalized data were first log2-transformed (log2(x + 1)). Proteins expressed at different levels between groups (downregulated proteins in green and upregulated proteins in red) were identified by a *P* value of <0.05 obtained from LIMMA's moderated *t* statistic (MAPK14, *\*p* = 0.0107). **b** Representative images of immunohistochemical stains for MAPK14, SMAD1/5/9 phosphorylation, and SMAD1 in formalin-fixed paraffin-embedded tissue sections from 344SQP and 344SQR tumors treated with IgG (10 mg/kg). Scale bar, 100 μm (×40 magnification). Data shown are representative of two reproducible independent experiments of three biologically independent samples. **c**, Representative images of immunohistochemical stains for MAPK14, SMAD1/5/9 phosphorylation, and SMAD1 in formalin-fixed paraffin-embedded tissue sections from BMP7-knockdown tumors treated with IgG (10 mg/kg) compared with control (scale bar, 100 μm) (×40 magnification). Data shown are representative of two reproducible independent experiments of three biologically independent samples. **d** Representative images of immunohistochemical stains for MAPK14, SMAD1/5/9 phosphorylation, and SMAD1 in formalin-fixed paraffin-embedded tissue sections from 344SQP and 344SQR tumors treated with anti-PD1 (10 mg/kg). Scale bar, 100 μm (×40 magnification). Data shown are representative of two reproducible independent experiments of three biologically independent samples. **e** Representative images of immunohistochemical stains for MAPK14, SMAD1/5/9 phosphorylation, and SMAD1 in formalin-fixed paraffin-embedded tissue sections from BMP7-knockdown tumors treated with anti-PD1(10 mg/kg) compared with control (scale bar, 100 μm) (×40 magnification). Data shown are representative of two reproducible independent experiments of three biologically independent samples. **f** Representative images of immunohistochemical stains for MAPK14 and SMAD1/5/9 phosphorylation in formalin-fixed paraffin-embedded tissue sections from two patients collected before treatment and at the time of disease progression on pembrolizumab or ipilimumab. Scale bar, 100 μm (×40 magnification). Data shown are representative of two reproducible independent experiments. **g** Nanostring immune panel results for 770 genes in tumor-infiltrating leukocytes (TILs) collected from 344SQP (*n* = 2 biologically independent samples) and 344SQR (*n* = 3 biologically independent samples) tumors treated with anti-PD1(10 mg/kg). Genes expressed at different levels between groups (downregulated genes in green and upregulated genes in red) were identified by a *P* value of <0.05 obtained from LIMMA's moderated *t* statistic (MAPK14, *p* = 0.0047). **h** Immunofluorescence analysis of MAPK14 and SMAD1/5/9 phosphorylation (green) in the macrophage cell line RAW 264.7 at 24 h after treatment with BMP7 (250 ng) or BMP7 plus follistatin (foll) (250 ng). DAPI (blue) was used to stain cellular nucleus. Data shown are representative of three reproducible independent experiments. Scale bar, 100 μm (×40 magnification).

(RANTES), IFNG, and IL2 (also related to MAPK14 signaling) were downregulated in serum from mice bearing 344SQR tumors versus parental tumors, although those findings were not evident in the Nanostring data. As BMP7 was shown to be correlated with TGFB, we analyzed serum levels of TGFG1, TGFG2, and TGFB3 from mice bearing 344SQR or 344SQP tumors treated with IgG control or anti-PD1 antibodies by enzyme-linked immunosorbent assay (ELISA). We did not find differences in TGFB levels in 344SQP vs. 344SQR treated with IgG control or 344SQP vs. 344SQR treated with anti-PD1 antibodies. Next, we evaluated whether BMP7 promotes MAPK14 downregulation via SMAD1 in TILs, as was previously seen for 344SQR tumors and patients' samples. We found that MAPK14 expression was higher in a macrophage cell line (RAW 264.7) treated with BMP7 plus the

BMP antagonist follistatin compared with BMP7 alone (Fig. 2h), and we found that p-SMAD1/5/9 was lower in cells treated with BMP7 plus follistatin versus BMP7 alone (Fig. 2h). These results suggest that BMP7 regulates MAPK14 expression via SMAD1 signaling not only in tumors resistant to anti-PD1 but also in immune cells.

## BMP7 reduces proinflammatory signaling via MAPK14 suppression.
In order to determine whether MAPK14 downregulation in TILs depended on BMP7 secretion in the tumor microenvironment, we evaluated the expression of *MAPK14* and *MAPK14*-regulated cytokines and chemokines in TILs isolated from BMP7-knockdown tumors and control tumors treated with IgG or

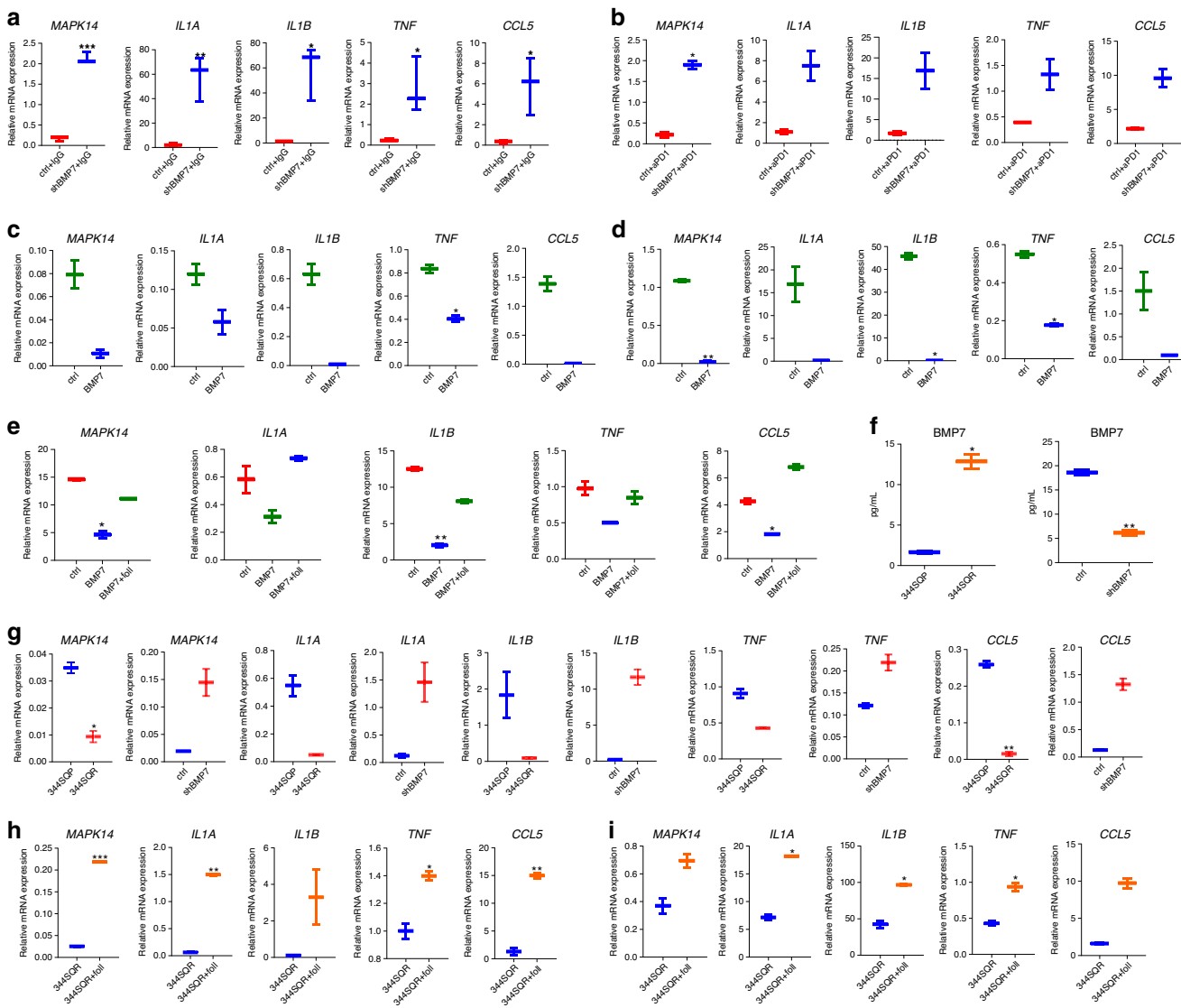

**Fig. 3 BMP7 reduced macrophage-mediated proinflammatory signaling via MAPK14. a**, **b** BMP7-knockdown and –control cells (0.5 × 10⁶) were injected into 129 Sv/Ev mice and treated with IgG (*n* = 3 biologically independent samples) or anti-PD1 (10 mg/Kg) (*n* = 2 biologically independent samples) twice a week for 2 weeks. A week after the final IgG and anti-PD1 treatment, tumor-infiltrating leukocytes (TILs) were collected, and expression of *MAPK14*, *IL1A*, *IL1B*, *TNF*, and *CCL5* were analyzed by quantitative PCR. Box-and-whisker plots show the minimum and maximum values. *P* values are from unpaired, two-sided *t* tests. Statistical significance was defined as *,*P* < 0.05, **,*P* < 0.01, ***,*P* < 0.001, and ****, *P* < 0.0001. **c**, **d** Quantitative PCR of *MAPK14*, *IL1A*, *IL1B*, *TNF*, and *CCL5* expression in RAW 264.7 cells **c** and peritoneal macrophages **d** untreated or treated with BMP7 (250 ng) for 24 or 48 h. Box-and-whisker plots show the minimum and maximum values of two independent experiments. **e** Quantitative PCR of *MAPK14*, *IL1A*, *IL1B*, *TNF*, and *CCL5* expression in RAW 264.7 cells untreated and treated with BMP7 (250 ng) or BMP7 plus follistatin (foll) (250 ng) for 24 h. Box-and-whisker plots show the minimum and maximum values of  two independent experiments. **f** BMP7 levels in cell culture supernatant from 344SQP, 344SQR, and 344SQR ctrl and 344SQR shBMP7 cells analyzed by enzyme-linked immunosorbent assay. Box-and-whisker plots show the minimum and maximum values of two biologically independent samples. **g** Quantitative PCR of *MAPK14*, *IL1A*, *IL1B*, *TNF*, and *CCL5* expression in RAW 264.7 cells co-cultured with 344SQP or 344SQR, and 344SQR shBMP7 or 344SQR ctrl cells for 24 h. Box-and-whisker plots show the minimum and maximum values of two independent experiments. **h**, **i** Quantitative PCR of *MAPK14*, *IL1A*, *IL1B*, *TNF*, and *CCL5* expression in RAW 264.7 cells and peritoneal macrophages co-cultured with 344SQR cells or 344SQR cells plus follistatin (foll) (250 ng) for 24 or 48 h. *CD45* expression was used as a housekeeping gene for quantitative PCR analysis. The comparative Ct method was used to calculate the relative abundance of mRNAs compared with *CD45* expression. Box-and-whisker plots show the minimum and maximum values of two independent experiments. *P* values are from unpaired, two-sided *t* tests. Statistical significance was defined as *,*P* < 0.05, **,*P* < 0.01, ***,*P* < 0.001, and ****,*P* < 0.0001.

anti-PD1. We found that *MAPK14*, *IL1A*, *IL1B*, *TNF*, and *CCL5* expression levels were increased in TILs from BMP7-knockdown tumors versus control (Fig. 3a, b; Supplementary Data 2). We silenced *MAPK14* in RAW 264.7 cells with siRNAs and analyzed *IL1A*, *IL1B*, *TNF*, and *CCL5* expression to confirm that *MAPK14* regulates *IL1A*, *IL1B*, *TNF*, and *CCL5* expression in RAW 264.7 cells (Supplementary Fig. 4a; Supplementary Methods). We then treated

RAW 264.7 cells and peritoneal macrophages with BMP7 and found that these cells had lower expression of *MAPK14* and *MAPK14*-regulated cytokines and chemokines when treated with BM7 versus untreated control (Fig. 3c, d). We next treated RAW 264.7 cells with BMP7 with or without follistatin and found that RAW 264.7 cells had lower expression of *MAPK14* and *MAPK14*-regulated cytokines and chemokines when treated with BM7 versus untreated control

and BM7 plus follistatin (Fig. 3e). We next investigated if tumor-secreted BMP7 regulates *IL1A, IL1B, TNF*, and *CCL5* via *MAPK14* in macrophages. We first measured BMP7 levels in media from 344SQP vs. 344SQR, and 344SQR ctrl vs. 344SQR shBMP7. As expected, 344SQR cells secreted higher levels of BMP7 than 344SQP cells, and BMP7-knockdown cells secreted lower BMP7 levels than 344SQR ctrl (Fig. 3f). We then co-cultured RAW 264.7 cells with 344SQP or 344SQR and found that macrophages cultured with 344SQR cells had lower expression of *MAPK14* and *MAPK14*-regulated cytokines and chemokines compared with cells co-cultured with 344SQP cells. To confirm that these findings depended on BMP7 rather than some other secreted molecule, we co-cultured macrophages with 344SQR ctrl or 344SQR shBMP7 cells. Macrophages co-cultured with 344SQR shBMP7 cells had higher expression of *MAPK14* and *MAPK14*-regulated cytokines and chemokines compared with 344SQR ctrl (Fig. 3g). We finally co-cultured RAW 264.7 cells and peritoneal macrophages with 344SQR cells and treated them with follistatin. We found that macrophages co-cultured with 344SQR and treated with follistatin or the BMP receptor inhibitor K02288 had higher expression of *MAPK14* and *MAPK14*-regulated cytokines and chemokines versus 344SQR co-culture only (Fig. 3h, i; Supplementary Fig. 5; Supplementary Methods). These findings demonstrate that BMP7 regulates proinflammatory cytokine and chemokine expression via *MAPK14* in macrophages.

**BMP7 regulates IFNG and IL2 via MAPK14 in CD4⁺ T cells.** As we found that IFNG and IL2 levels were decreased in serum from mice bearing 344SQR tumors versus parental tumors, we next tested if BMP7 affected the expression of *IFNG* and *IL2* in T cells via *MAPK14*. Because we previously observed that BMP7 led to changes in *MAPK14* expression in CD4⁺ T cells but not in CD8⁺ T cells, we focused here on CD4⁺ T cells. First, to investigate whether BMP7 promotes MAPK14 downregulation via SMAD1 in CD4⁺ T cells, we cultured CD4⁺ T cells and treated them with BMP7 with or without follistatin for 60 min and evaluated *MAPK14* expression and SMAD1/5/9 activation via western blotting. *MAPK14* expression was higher in CD4⁺ T cells treated for 60 min with BMP7 plus follistatin versus BMP7 alone compared with actin protein levels (Fig. 4a). On the other hand, SMAD1/5/9 activation was lower in CD4⁺ T cells treated for 60 min with BMP7 plus follistatin versus BMP7 alone compared with actin protein levels as previously published by different studies[26,27] (Fig. 4a; Supplementary Fig. 11). These findings suggest that BMP7 negatively regulates *MAPK14* expression via SMAD1 signaling not only in tumors and macrophages but also in CD4⁺ T cells. We next investigated *IFNG* and *IL2* expression in TILs isolated from BMP7-knockdown tumors as compared with control tumors treated with IgG or anti-PD1, and we found that *IFNG* and *IL2* expression levels were increased in TILs from BMP7-knockdown tumors versus control (Fig. 4b, c). In order to confirm that *MAPK14* regulates *IFNG* and *IL2* expression, we silenced *MAPK14* in EL4 T cells with shRNAs and analyzed these cytokines expression via quantitative PCR (Supplementary Fig. 4b; Supplementary Methods). We next co-cultured CD4⁺ T cells with 344SQP or 344SQR cells, and 344SQR ctrl or 344SQR shBMP7 cells, and analyzed the expression of *MAPK14, IFNG*, and *IL2*. For these experiments, we harvested mouse spleens, collected CD4⁺ T cells by using magnetic beads, and activated those cells with CD3/CD28 antibodies before treatment. We found that co-culture of activated CD4⁺ T cells with 344SQR cells led to decreased *MAPK14, IFNG*, and *IL2* expression compared with 344SQP cells (Fig. 4d). To confirm that these findings depended on BMP7 and not on some other secreted molecule, we co-cultured CD4⁺ T cells with 344SQR ctrl or 344SQR shBMP7 cells. Co-culture of CD4⁺

T cells with 344SQR shBMP7 upregulated *MAPK14, IFNG*, and *IL2* expression compared with 344SQR ctrl (Fig. 4e). We then treated CD4⁺ T cells with BMP7 with or without follistatin and evaluated *MAPK14, IFNG*, and *IL2* expression. We found that CD4⁺ T cells had lower expression of *MAPK14, IFNG*, and *IL2* when treated with BMP7 compared with untreated control and BMP7 plus follistatin (Fig. 4f). We next co-cultured CD4⁺ T cells with 344SQR cells and treated them with follistatin and found that those CD4⁺ T cells had higher expression of *MAPK14, IFNG*, and *IL2* versus 344SQR without follistatin (Fig. 4g). These findings suggest that BMP7 negatively regulates *IFNG* and *IL2* expression via SMAD1/MAPK14 in CD4⁺ T cells.

**BMP7 inhibition re-sensitizes anti-PD1-resistant tumors.** Next, we tested if BMP7 knockdown could sensitize anti-PD1-resistant tumors to immunotherapy. We injected 344SQR ctrl and 344SQR shBMP7 cells into 129 Sv/Ev mice and treated the mice with IgG control or anti-PD1. BMP7 knockdown was found to re-sensitize tumors to anti-PD1 and extended mouse survival relative to the control group (Fig. 5a). We then evaluated if BMP7 inhibition via follistatin could re-sensitize resistant tumors. We found that BMP7 inhibition via follistatin decreased tumor growth and extended survival compared with anti-PD1 therapy only (Fig. 5b). In order to confirm these effects are not specific to 344SQR cell line, we tested if BMP7 knockdown in 4T1 mouse mammary carcinoma model promote sensitivity to anti-PD1 therapy. Previous studies have shown that 4T1 is poorly immunogenic and 4T1-derived tumors could not generally be cured by anti-PD1 or anti-CTLA-4[28]. We first established 4T1 BMP7-knockdown tumors and control cell lines and then we injected these cells in BALB/c mice and treated tumors with IgG or anti-PD1. Our results demonstrated that BMP7-knockdown sensitized 4T1 mouse mammary carcinoma tumors to anti-PD1 therapy and extended mouse survival relative to the control group (Supplementary Fig. 6; Supplementary Methods). We found increased percentages of CD8⁺ T cells in the BMP7-knockdown tumors treated with anti-PD1 versus BMP7-knockdown tumors treated with IgG or control tumors treated with IgG or anti-PD1 (Fig. 5c). We also found more CD8⁺ IFNG⁺ T cells in BMP7-knockdown tumors treated with anti-PD1 or IgG than in control tumors treated with IgG (Fig. 5c; Supplementary Fig. 7a). Next, we evaluated the percentages of M2 macrophages (CD206 marker) in BMP7-knockdown tumors treated with anti-PD1, and found that BMP7-knockdown tumors treated with IgG or anti-PD1 had decreased percentages of M2 macrophages compared with control tumors treated with IgG or anti-PD1 (Fig. 5d). We then evaluated percentages and activation of CD4⁺ T cells (via IFNG production) in BMP7-knockdown tumors treated with anti-PD1. We found an increase in percentage of CD4⁺ T cells in BMP7-knockdown tumors treated with IgG or anti-PD1 compared with control tumors treated with IgG or anti-PD1 (Fig. 5e). We also found more CD4⁺ IFNG⁺ T cells in BMP7-knockdown tumors treated with IgG or anti-PD1 than in control tumors (Fig. 5e; Supplementary Fig. 7b). We then evaluated M2 macrophage and CD4⁺ T-cell infiltration by mmunohistochemical (IHC) staining. Concordant with the flow cytometry data, we found that infiltration of M2 macrophages was decreased in BMP7-knockdown tumors treated with IgG or anti-PD1 compared with control tumors (Fig. 5f and Supplementary Fig. 8). On the other hand, we found that infiltration of CD4⁺ T cells was higher in BMP7-knockdown tumors treated with IgG or anti-PD1 compared with control tumors (Fig. 5g). We also tested the combination of BMP7 knockdown and anti-CTLA4 or anti-PDL1. We found that antibodies to both PDL1 and CTLA4 increased survival in combination with BMP7-knockdown compared with control (Fig. 5h, i). In

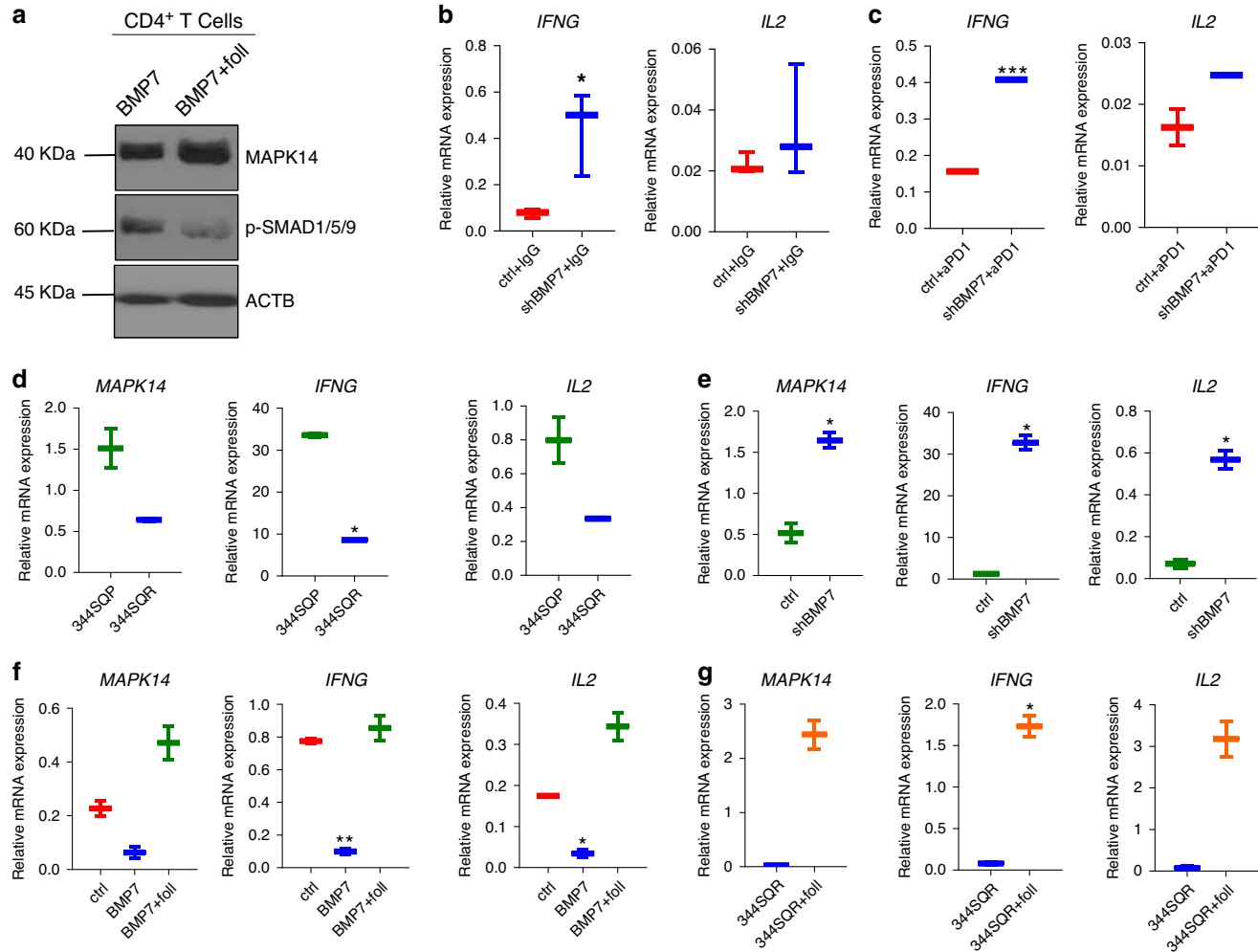

**Fig. 4 BMP7 regulates CD4+ T-cell production of IFNG and IL2 via MAPK14. a** Western blotting analysis of MAPK14 and SMAD1/5/9 phosphorylation in CD4+ T cells at 60 min after treatment with BMP7 (250 ng) or BMP7 plus follistatin (foll) (250 ng). ACTB expression was used for normalization in western blotting. **b**, **c** BMP7-knockdown and –control cells (0.5 × 10$^6$) were injected into 129 Sv/Ev mice and treated with IgG (n = 3 biologically independent samples) anti-PD1 (10 mg/Kg) (n = 2 biologically independent samples) twice a week for 2 weeks. A week after the final IgG or anti-PD1 treatment, tumor-infiltrating leukocytes (TILs) were collected, and expression of *IFNG*, and *IL2* were analyzed by quantitative PCR. Box-and-whisker plots show the minimum and maximum values. *P* values are from unpaired, two-sided *t* tests. Statistical significance was defined as *$P < 0.05$, **$P < 0.01$, ***$P < 0.001$, and ****$P < 0.0001$. **d**, **e** Quantitative PCR of *MAPK14*, *IFNG*, and *IL2* expression in CD4 + T cells co-cultured with 344SQP or 344SQR, and 344SQR shBMP7 or 344SQR ctrl cells for 24 h. Box-and-whisker plots show the minimum and maximum values  of two independent experiments. **f** Quantitative PCR of *MAPK14*, *IFNG*, and *IL2* expression in CD4+ T cells untreated and treated with BMP7 (250 ng) or BMP7 plus follistatin (foll) (250 ng) for 24 h. Box-and-whisker plots show the minimum and maximum values of two independent experiments. **g** Quantitative of *MAPK14*, *IFNG*, and *IL2* expression in CD4+ T cells co-cultured with 344SQR cells or 344SQR cells plus follistatin (foll) (250 ng) for 24 h. Box-and-whisker plots show the minimum and maximum values of two independent experiments. *CD45* expression was used as a housekeeping gene for quantitative PCR analysis. The comparative Ct method was used to calculate the relative abundance of mRNAs compared with *CD45* expression. *P* values are from unpaired, two-sided *t* tests. Statistical significance was defined as *$P < 0.05$, **$P < 0.01$, ***$P < 0.001$, and ****$P < 0.0001$.

order to determine whether anti-PD1 response in BMP7-knockdown tumors is T-cell or macrophage dependent, we performed depletion experiments in BMP7-knockdown tumors treated with anti-PD1. As shown on Supplementary Fig. 9, CD4+ T cells depletion completely reverted anti-PD1 response seen in BMP7-knockdown treated tumors. There was no difference between tumors treated with anti-PD1 alone or in combination with macrophages depleting antibody (Supplementary Fig. 9; Supplementary methods). Finally, we evaluate the prognostic significance of BMP7 expression in samples from 127 patients with NSCLC from The Cancer Genome Atlas (TCGA)[29]. In TCGA lung adenocarcinoma cohort, we found BMP7 significant in univariate Cox analysis but not significant in multivariate Cox model including BMP7 and Stage (Supplementary Table 5a, b). As

the *p* value of BMP7 is close to significance in the multivariate model, we searched in The Gene Expression Omnibus (GEO) database for a second cohort of lung adenocarcinoma patients to have at least 100 patients. We found GSE50081 with 181 Stage I and II NSCLC cases from Der SD et al. 2014[30]. We retrieved microarray (Affymetrix Human Genome U133 Plus 2.0 Array) expression (normalized log2) data for BMP7 along with clinical information for the patients. Among them, 127 cases are adenocarcinoma cancers. We performed univariate and multivariate Cox analysis and BMP7 (21160_at) was an independent marker of poor overall survival (Supplementary Table 5c–e, Supplementary Fig. 10a; Supplementary methods). We also analyzed the correlation between BMP7 expression and immune cells markers, including MAPK14, CD68 (resident-tissue macrophages maker),

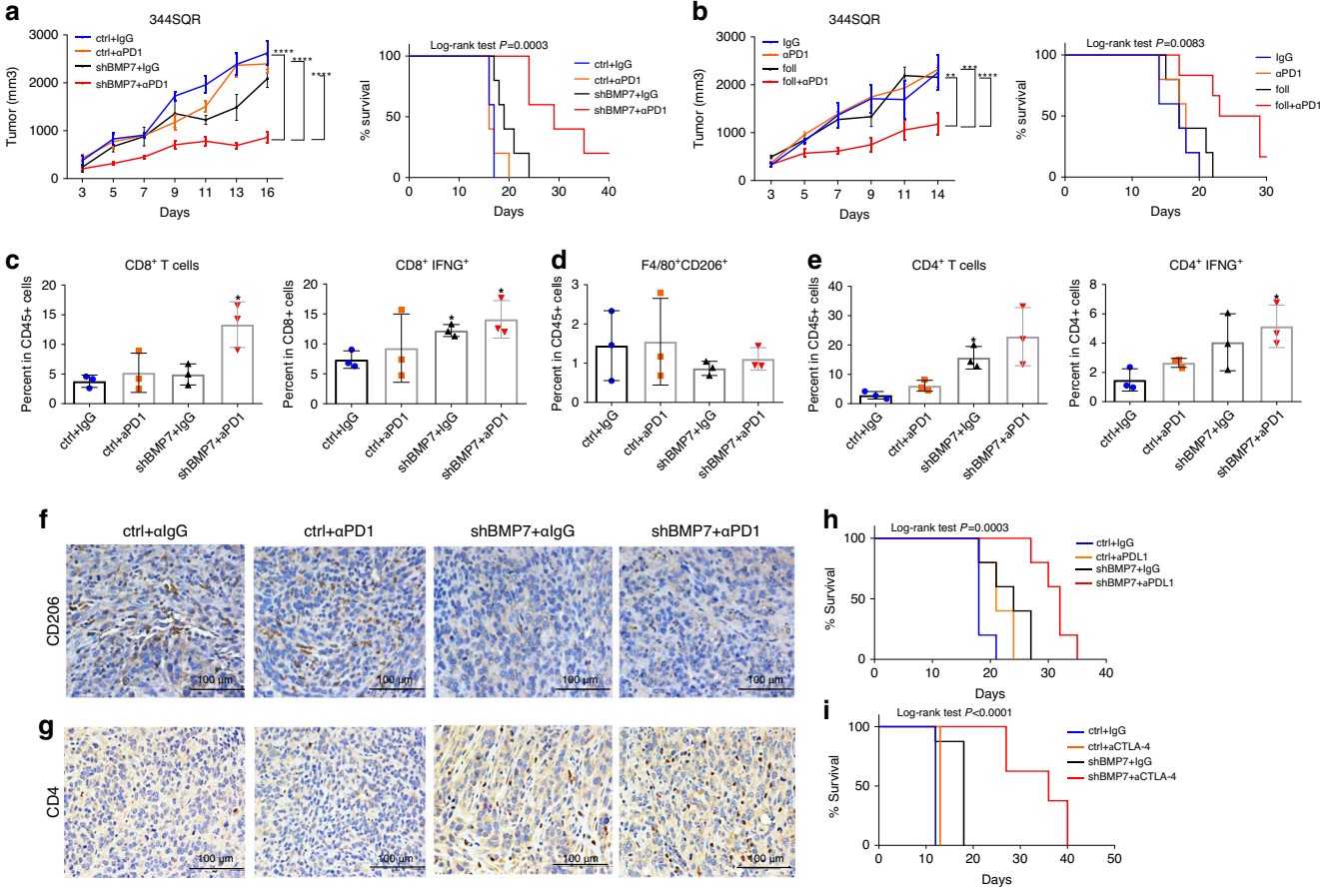

**Fig. 5 BMP7 inhibition re-sensitizes resistant tumors to anti-PD1 therapy. a** Tumor growth and survival analysis of mice with 344SQR tumors treated with IgG ctrl ($n = 5$ animals) or anti-PD1 (10 mg/kg) ($n = 5$ animals) or 344SQR shBMP7 tumors treated with IgG ($n = 5$ animals) or anti-PD1 (10 mg/kg) ($n = 5$ animals) twice a week for 2 weeks. **b** Tumor growth and survival analysis of mice with 344SQR tumors ($n = 5$ animals) treated with IgG, anti-PD1 (10 mg/kg), follistatin (0.1 mg/kg), or follistatin (0.1 mg/kg) plus anti-PD1(10 mg/kg) for 2 weeks. For a, ctrl+ IgG vs. shBMP7 + αPD1, ****$p < 0.0001$, ctrl+ αPD1 vs. shBMP7 + αPD1, ****$p < 0.0001$ shBMP7+IgG vs. shBMP7 + αPD1, ****$p < 0.0001$, Two-way RM ANOVA. For b, IgG vs. foll+αPD1, **$p = 0.0060$, αPD1 vs. foll+αPD1, ***$p = 0.0003$, foll+IgG vs. foll+αPD1, ****$p < 0.0001$, Two-way RM ANOVA. Statistical significance was defined as *$P < 0.05$, **$P < 0.01$, ***$P < 0.001$, and ****$P < 0.0001$. Mouse survival rates were analyzed by the Kaplan–Meier method and compared with log-rank tests. **c–e** Flow cytometry analysis of CD8+(*$p = 0.0421$), CD8+IFNG+(*$p = 0.0475$, *$p = 0.0121$), F4/80+CD206+, CD4+ (*$p = 0.0199$), CD4+IFNG+ T cells (*$p = 0.0303$) in tumor-infiltrating leukocytes (TILs) from 344SQR ctrl ($n = 3$ biologically independent samples) and 344SQR shBMP7 ($n = 3$ biologically independent samples) tumors treated with IgG or anti-PD1 (10 mg/kg) twice a week for 2 weeks. Data are presented as mean values ±SD. $P$ values are from unpaired, two-sided $t$ tests. **f, g** Representative images of immunohistochemical stains for CD206 (M2 macrophage marker) and CD4 (brown dots) in formalin-fixed paraffin-embedded tissue sections from BMP7-knockdown tumors treated with IgG or anti-PD1 compared with control. A representative staining image from each cohort ($n = 3$ biologically independent samples) is displayed. Data shown are representative of two reproducible independent experiments. Scale bar, 100 μm (×40 magnification). **h, i** Survival analysis of mice with 344SQR ctrl tumors or 344SQR shBMP7 tumors treated with IgG or anti-PDL1 (10 mg/kg) ($n = 5$ animals) or anti-CTLA4 (10 mg/kg) ($n = 8$ animals) twice a week for 2 weeks. Mouse survival rates were analyzed with the Kaplan–Meier method, and curves compared with log-rank tests. Statistical significance was defined as *$P < 0.05$, **$P < 0.01$, ***$P < 0.001$, and ****$P < 0.0001$.

FOXP3 (T regulatory cells marker), CD8A, and CD3E in samples from 127 patients with NSCLC from GSE50081[30]. We found an inverse correlation between BMP7 and MAPK14 and a positive correlation with CD68 and FOXP3 in patients with NSCLC (Supplementary Fig. 10b; Supplementary Methods). Collectively, these findings suggest that BMP7 inhibition or treatment with follistatin may represent a potential therapeutic approach to overcome resistance to immunotherapies such as anti-PD1, anti-CTLA4, and anti-PDL1.

## Discussion

In the current study, we identify BMP7 as a regulator of resistance to immunotherapies. BMPs are secreted proteins that belong to the TGF-β superfamily and regulate proliferation, differentiation, and apoptosis in many different cell types, including immune cells. Binding of BMP to its receptor leads to the phosphorylation of intracellular Smads, which then bind to co-Smad 4 and translocate into the nucleus to regulate gene expression. BMP7 was upregulated in mouse and human tumors resistant to anti-PD1 therapy, and BMP7 levels were higher not only in blood from mice bearing resistant tumors but also in pretreatment blood from patients exhibiting disease progression while on anti-PD1 and radiotherapy. Our studies reveal that secreted BMP7 impinges on effector T-cell functions, favoring the generation of immunosuppressive cells, and precluding response to immunotherapy. Collectively, our results suggest that targeting BMP7 may represent a therapeutic approach to overcome resistance to checkpoint blockade therapies in cancer.

The BMP7 promoter was hypomethylated in anti-PD1-resistant tumors from our preclinical model, which explains its upregulated RNA and protein levels. Previous studies have also shown that BMP7 expression can be regulated via epigenetic mechanisms[31,32]. Interestingly, others have shown that epigenetic drugs targeting histone deacetylation or methylation modulate the immune response and overcome acquired resistance to immunotherapy. For example, epigenetic drugs enhance the efficacy of immune checkpoint inhibitor therapy by increasing the expression of immune checkpoint ligands and tumor-associated antigens on tumor cells[33]. Our results may suggest that epigenetics drugs can also modulate genes that promote immunosuppression such as BMP7. Nonetheless, further studies in a large number of human derived tumors are necessary to confirm this hypothesis. In addition, further studies are also needed to determine the mechanism behind BMP7 upregulation in tumors and blood from patients with disease progression while on immunotherapies.

Along with BMP7, we found other genes to be epigenetically modulated in the anti-PD1-resistant tumors compared with parental tumors. The hypomethylated genes included Klhl1, Slc2a13, Snord37, and Fam196 and hypermethylated included Nav1, Palm3, and Clic6. To date, we did not find studies correlating these genes with resistance to immunotherapies and its role in acquired resistance to anti-PD1 is currently under investigation in our laboratory.

Our studies revealed that proteins known to be related to BMP7 signaling, such as MAPK14[24,25,34–37], were expressed differently in resistant tumors than in parental tumors. Because MAPK14 is known to be regulated by BMP7[23,24] via SMAD activation[25], we focused on this protein for validation studies. BMP7 can either promote or inhibit MAPK14 activation depending on the cellular context and BMP7 dose[23–25,36]. Here, we found that BMP7 specifically regulated MAPK14 at the mRNA and protein levels. We then validated MAPK14 downregulation in 344SQP versus 344SQR tumors, and in cancer patients with progression on immunotherapies. We also validated SMAD1 activation status in patient samples with high BMP7 expression by IHC analysis. These findings suggest that BMP7 downregulates MAPK14 via SMAD1 activation in tumors resistant to anti-PD1 therapy. MAPK14 can act as a tumor suppressor by regulating cell cycle progression and induction of apoptosis or as an oncogene by promoting invasion, inflammation, and angiogenesis[38]. Although our goal in this study was to study the effect of BMP7 on immune cells in the tumor microenvironment, the downregulation of MAPK14 by BMP7 in resistant tumors cells might also be an important mechanism of resistance to immunotherapies that deserve further investigation.

MAPK14 is a member of the p38 MAPK family and it was downregulated not only in tumors but also in TILs from anti-PD1-resistant tumors. P38 proteins are important participants in inflammatory signaling pathways and are activated in response to a variety of cellular stresses, including osmotic shock, lipopolysaccharides, and inflammatory cytokines[39–43]. MAPK14 is the critical isoform in inflammatory responses and is involved in the expression of proinflammatory mediators in macrophages such as IL1B, TNF, and IL12[44–46] as well as CCL5[47], COX-2, IL8, IL6, IL3, IL2, and IL1, all of which contain AU-rich elements in their 3′-untranslated regions to which MAPK14 binds[48]. MAPK14 participates in the regulation of IFNG expression and its mRNA stabilization in immune cells[49,50]. Strikingly, the MAPK14-regulated inflammatory cytokines IL1A, IL1B, and TNF were downregulated in TILs collected from 344SQR tumors treated with anti-PD1 versus 344SQP. Cytokines and chemokines regulated by MAPK14 including IL1A, IL1B, TNF, CCL5, IFNG, and IL2 were also downregulated in blood from mice bearing 344SQR tumors

compared with parental tumors. Others have also found that BMP7 treatment led to significant reductions in proinflammatory cytokines, including TNF, in macrophages in vivo[20,51] and that BMP7 represses TNF and IL1B in models of chronic and acute renal failure and in chondrocytes from patients with osteoarthritis[52,53]. Our findings confirmed that MAPK14, IL1A, IL1B, TNF, CCL5, IFNG, and IL2 expression levels were increased in TILs isolated from BMP7-knockdown tumors. These results suggest that BMP7 regulates MAPK14 expression not only in tumors resistant to anti-PD1 but also in TILs in the tumor microenvironment, and that BMP7 also regulates expression of proinflammatory cytokines and chemokines in TILs via MAPK14 regulation. Next, to investigate whether BMP7 regulates MAPK14 via SMAD1 activation in immune cells, as we saw in tumors, we treated macrophages and CD4+ T cells with BMP7, with or without its natural inhibitor follistatin, and analyzed SMAD1/5/9 activation. Our results suggest that BMP7 also regulates MAPK14 through SMAD1 activation in these cells. These findings are supported by previous studies in macrophages isolated from a vivo model of atherosclerosis treated with intravenous injections of BMP7 or liposomal clodronate[51]. In that study, BMP7 significantly reduced the number of proinflammatory macrophages and decreased MAPK14 activation while increasing SMAD1/5/8 phosphorylation in macrophages. Other studies have shown that BMP7 promotes M2 polarization in human and mouse macrophages in vitro and in vivo models[20–22].

MAPK14 signaling promotes not only M2 monocytes polarization into M1-type cells in response to lipopolysaccharides[54] but also is central in the activation of proinflammatory gene transcription. In macrophages, MAPK14 is activated by lipopolysaccharide and Toll-like receptor-4, which subsequently activates proinflammatory cytokines, including IL1 and TNF[39–42]. Therefore, we next investigated whether secreted BMP7 reduced proinflammatory and chemokines via MAPK14 in macrophages. Of note, BMP7 physiological levels were not directly measured to define the dose in our experiments. We used a concentration of 250 ng BMP7 for our in vitro studies based on previous studies that tested different concentrations that promoted SMAD1/5/9 activation in macrophages[22]. We found that murine macrophages co-cultured with 344SQR cells had lower expression of MAPK14, IL1A, IL1B, TNF, and CCL5 compared with cells co-cultured with 344SQP cells. We confirmed that these findings depended on BMP7 by co-culturing macrophages with BMP7-knockdown 344SQR cells. We found that murine macrophages co-cultured with BMP7-knockdown 344SQR cells expressed higher levels of MAPK14, IL1A, IL1B, TNF, and CCL5 compared with 344SQR ctrl cells. We then treated RAW 264.7 cells with BMP7, with or without follistatin. As expected, these cells expressed lower levels of MAPK14, IL1A, IL1B, TNF, and CCL5 when treated with BMP7 compared with untreated cells or cells treated with BMP7 plus follistatin. Taken together, these findings show that BMP7 suppresses the proinflammatory cytokine expression regulated by MAPK14 in macrophages.

P38 signaling is known to be activated in T cells stimulated via TCR signaling and reduced in anergic T cells[55]. MAPK14 also participates in the regulation of IFNG expression in CD4+ T cells[49] and promotes the 3¢-untranslated region stabilization of IFNG mRNA in NK cells[50]. Further, the inhibition of MAPK14 in Th1 cells differentiated in vitro blocked the IFNG expression induced by IL12/IL18 and CD3/CD28 stimulation[56–58]. Previous studies showed that treating cells with SB203580, a specific inhibitor of MAPK14, suppressed the transcriptional activation of the IL2 promoter in T lymphocytes[59,60]. Therefore, we investigated the effect of BMP7 on IFNG and IL2 in T cells. SMAD regulatory pathways regulate different aspects of immune activation and immune suppression in T cells[61]. For example, TGFB

promotes the differentiation of CD4$^+$ T cells into suppressive FOXP3$^+$ T regulatory cells via SMAD activation[62]. In the current study, our findings suggest that BMP7 regulates MAPK14 expression via SMAD1 signaling not only in tumors and macrophages but also in CD4$^+$ T cells. Activated CD4$^+$ T cells incubated with BMP7 had lower MAPK14, IFNG, and IL2 expression compared with untreated cells and BMP7 plus follistatin-treated cells. In agreement with these findings, other studies have also correlated IL2 activation with MAPK14 signaling in T cells[63–65]. Notably, other BMPs can promote or inhibit T-cell proliferation and IFNG and IL2 production[16]. Indeed, BMP2, BMP4, and BMP6 can promote CD4$^+$ T-cell proliferation and IL2 production[66]. Nonetheless, the effect of BMP7 on CD4$^+$ T cells was not clear. In this study, we found that BMP7 decreased IFNG and IL2 expression in CD4$^+$ T cells via SMAD1/MAPK14 signaling.

We then tested whether BMP7 knockdown could re-sensitize anti-PD1-resistant tumors to immunotherapy. We found that BMP7 knockdown in anti-PD1-resistant 344SQR model and in 4T1 mouse mammary carcinoma model sensitized tumors to anti-PD1 and extended survival relative to the control. BMP7 pharmacological neutralization via follistatin re-sensitized tumors to anti-PD1 and extended survival relative to the control. As follistatin not only neutralizes BMP7 but other members of the TGFB superfamily such as activins, it might represent a broader approach to overcome resistance to anti-PD1. Interestingly, the combination of BMP7 knockdown and anti-CTLA4 or anti-PDL1 also extended survival compared with control, which suggests that mechanisms of resistance to anti-PD1 might overlap with resistance to anti-CTLA4 or anti-PDL1. We further found increased numbers of CD4$^+$ T cells in BMP7-knockdown tumors treated with anti-PD1 or IgG compared with control. We also found that CD4$^+$ IFNG$^+$ T cells were higher in BMP7-knockdown tumors treated with anti-PD1 or IgG than in control tumors treated with IgG. We found increased numbers and activation of CD8$^+$ T cells in BMP7-knockdown tumors treated with anti-PD1. On the other hand, BMP7-knockdown tumors treated with IgG or anti-PD1 had decreased percentages of M2 macrophages compared with control tumors treated with IgG or anti-PD1. These findings are supported by others showing that BMP7 increases M2 macrophage differentiation in vitro and in vivo in different models[20–22,51]. We next sought to clarify if anti-PD1 response in BMP7-knockdown tumors was T-cell or macrophage dependent. To address this question, we performed depletion experiments using specific antibodies to deplete CD4$^+$ T cells or macrophages in BMP7-knockdown tumors treated with anti-PD1. Our results showed that CD4$^+$ T-cell depletion completely reverted anti-PD1 response seen in BMP7-knockdown treated tumors. There was no difference between tumors treated with anti-PD1 alone or in combination with macrophages depleting antibody. These results might be explained by the fact that M2 macrophages were also depleted which might improve antitumor immune response.

In conclusion, we demonstrated that secreted BMP7 promotes resistance to anti-PD1 therapy by repressing macrophage-mediated inflammatory responses and Th1-associated cytokines in the tumor microenvironment. BMP7 downregulated MAPK14 and MAPK14-regulated cytokines and chemokines including IL1A, IL1B, TNF, and CCL5 via SMAD1 activation. At the same time, BMP7 decreased CD4$^+$ T-cell activation by downregulating IFNG and IL2 expression via SMAD1/ MAPK14 signaling (Fig. 6). Collectively, these findings suggest that BMP7 inhibition may represent a target for overcoming resistance to cancer immunotherapies.

## Methods

**Patient tumor and blood samples.** Tumor biopsies from a NSCLC patient with disease progression after pembrolizumab (NCT02444741), a patient with

adrenocortical carcinoma and a patient with Mixed Mullerian carcinoma with disease progression after ipilimumab (NCT02239900) were examined. Paraffin-embedded tissues were used for the IHC analysis. Biopsies were collected prior treatment with radiotherapy. Pretreatment blood samples from patients with PD on pembrolizumab (NCT02444741; NCT02402920) ($n = 5$) versus patients with PR or SD ($n = 4$) were collected in ethylenediaminetetraacetic acid (EDTA) tubes. Blood samples were centrifuged at $1000 \times g$ for 10 min, and plasma samples were collected and kept at −80 °C until analysis. All patients provided written informed consent before enrollment and the study protocol and amendments were approved by The University of Texas MD Anderson. Cancer Center Institutional Review Board (protocols 2014-1020 and 2013-0882).

**Cell lines.** The 344SQ parental cell line (344SQP) was a generous gift from Dr. Jonathan Kurie (MD Anderson). From the 344SQP cell line, we generated an anti-PD1-resistant cell line (344SQR) as previously described[4]. RAW 264.7 and EL4 cell lines were obtained from the American Type Culture Collection (ATCC; Manassas, VA, USA). Cell lines were cultured in complete medium (RPMI-1640 supplemented with 100 units/mL penicillin, 100 μg/mL streptomycin, and 10% heat-inactivated fetal bovine serum) and incubated at 37 °C in 5% $CO_2$. Cell lines were validated by DDC Medical (http://ddcmedical.com; Fairfield, OH) by short-tandem-repeat DNA fingerprinting.

**Establishment of stable BMP7- and MAPK14-knockdown cells.** To establish stable BMP7-knockdown cells, GIPZ Non-silencing Lentiviral shRNA Control (Catalog #RHS4348, Dharmacon) and specific mouse shRNA targeting BMP7 (pGIPZ Clone ID V2LMM_12865, Dharmacon) and MAPK14 (pGIPZ Clone ID V3LMM_415230, Dharmacon) viral supernatants were purchased from the shRNA and ORFeome Core at MD Anderson Cancer Center. 344SQR and EL4 cells were infected and incubated with the viral particles supplemented with Polybrene (8 μg/ mL, Sigma) overnight at 37 °C. Puromycin (1 μg/mL) was used to select and maintain BMP7-knockdown in 344SQR cells and MAPK14-knockdown in EL4 cells. Stable repression was verified by quantitative PCR and western blotting.

**Reduced representation bisulfite sequencing.** Reduced representation bisulfite sequencing (RRBS) was done by the Epigenomics Profiling Core and Science Park NGS facility at MD Anderson Cancer Center. A KAPA Library Quantification Kit (KAPA Biosystems) was utilized to quantify RRBS libraries for pooling, and a final concentration of 1.5 nM was loaded onto an Illumina cBOT for cluster generation before sequencing on an Illumina HiSeq 3000 using a Single Read 50 bp run. The libraries were sequenced using 50 bases single read protocol on Illumina HiSeq 3000 instrument. 49–85 million reads were generated per sample. Mapping: The adapters were removed from 3¢ ends of the reads by Trim Galore! (version 0.4.1) (https://www.bioinformatics.babraham.ac.uk/projects/trim_galore/) and cutadapt (version 1.9.1). Then, the reads were mapped to mouse genome mm10 by the bisulfite converted read mapper Bismark (version v0.16.1) and Bowtie (version 1.1.2). In all, 92–94% reads were mapped to the mouse genome, with 66–68% uniquely mapped. In all, 33–56 million uniquely mapped reads were used in the final analysis. Methylation calling: The methylation percentages for CpG sites were calculated by the bismark_methylation_extractor script from Bismark and an in-house Perl script. Differential methylation: the differential methylation on CpG sites was statistically assessed by R/Bioconductor package methylKit (version 0.9.5). The CpG sites with read coverage ≥20 in all the samples were qualified for the test. The significance of differential methylation on gene level was calculated using Stouffer's z score method by combining all the qualified CpG sites inside each gene's promoter region (defined as −1000 bp to +500 of TSS), and was corrected to FDR by Benjamini & Hochberg method. Heatmap and clustering: heatmap and clustering were performed on the top 10 hypermethylated genes and top 10 hypomethylated genes (by FDR) from the genes with number of CpG sites in promoter ≥5 and methylation difference ≥20%. Hierarchical clustering was done by hclust function in R using the average methylation percentage of all the qualified CpG sites in each gene's promoter region. Before clustering, for each gene, the methylation percentages across samples were centered by median and rescaled so that the sum of the squares is 1.0. Euclidean distance and ward.D2 clustering method were used for the clustering of the genes. The heatmap was plotted by heatmap.2 function in R.

**Pyrosequencing methylation assay.** Bisulfite PCR was done by the Epigenomics Profiling Core Facility at MD Anderson Cancer Center[67]. Genomic DNA (2 μg) was denatured with 0.2 M NaOH at 37 °C for 10 min followed by incubation with 30 μL of freshly prepared 10 mM hydroquinone and 520 μL of 3 M sodium bisulfite (pH 5.0) at 50 °C for 16 h. DNA was purified on a Wizard Miniprep Column (Promega), desulfonated with 0.3 M NaOH at 25 °C for 5 min, precipitated with ammonium acetate and ethanol, and dissolved in 50 μL of Tris-EDTA buffer (10 mM Tris-HCl, 1 mM EDTA, pH 8.0). Bisulfite-treated DNA (40–80 ng) was amplified with gene-specific primers by PCR. The following PCR conditions were used: initial denaturation at 95 °C for 10 min, followed by 45 cycles comprising denaturation at 95 °C for 30 s, annealing at 55 °C for 30 s, and extension at 72 °C for 30 s. PCR products from the second step were cloned into the pCR4-TOPO vector (Invitrogen), transformed into competent bacteria, and sequenced.

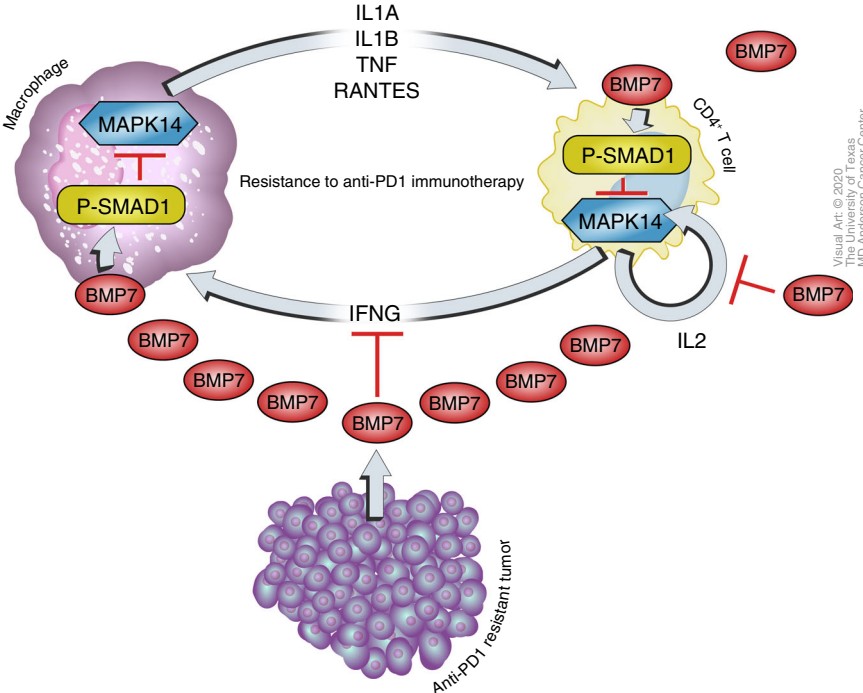

**Fig. 6 Work model.** Tumors with acquired resistance to immunotherapies secretes BMP7, a member of the TGFβ superfamily, repressing macrophage-mediated inflammatory responses and Th1-associated cytokines in the tumor microenvironment. BMP7 downregulated MAPK14 and MAPK14-regulated cytokines and chemokines including IL1A, IL1B, TNF, and CCL5 via SMAD1 activation in macrophages. At the same time, BMP7 decreased CD4⁺ T-cell activation by downregulating IFNG and IL2 expression via SMAD1/ MAPK14 signaling. Taken together, these findings suggest that BMP7 may represent a potential target for therapeutic approaches to overcome resistance to immunotherapies.

Sequences for the primers used, PCR reactions and cycling conditions in this study are provided in Supplementary Table 2. We measured levels of DNA methylation as the percentage of bisulfite-resistant cytosines at CpG sites by pyrosequencing with the PSQ HS 96 Pyrosequencing System (Biotage, Charlottesville, VA) and Pyro Gold CDT Reagents (Biotage).

**Enzyme-linked immunosorbent assay.** Serum was collected from mice bearing 344SQP or 344SQR tumors treated with anti-PD1 twice per week for a total of four doses. A week after the last anti-PD1 treatment, whole blood samples were collected by cardiac puncture and centrifuged at $1000 \times g$ for 10 min, and serum was collected and kept at –80 °C until analysis. Culture supernatants were freshly collected from 344SQP, 344SQP, 344SQR ctrl, and shBMP7 tumors and directly submitted for analysis. Plasma samples from patients with PD on pembrolizumab (NCT02444741; NCT02402920) and radiotherapy versus patients with PR or SD were collected as previously described. BMP7 levels in serum, plasma, or culture supernatants were measured by ELISA according to the manufacturer's protocol (Thermo Fisher Scientific, Catalog #EHBMP7 and EMBMP7).

**Quantitative polymerase chain reaction.** Total RNA was isolated from cells and tumors with Triazol (Life Technologies) according to the manufacturer's protocol. For studies of BMP7, MAPK14, IL1A, IL1B, TNF, CCL5, IL2, and IFNG expression, mRNA was retrotranscribed with the iScript™ gDNA Clear cDNA Synthesis Kit (BioRad) and analyzed by quantitative PCR using SYBR Green (Life Technologies) with specific primers (Supplementary Table 6) according to the manufacturer's protocol. The comparative Ct method was used to calculate the relative abundance of mRNAs compared with *ACTB* (beta-actin) expression for cancer cells or *CD45* expression for immune cells.

**IHC analysis.** Formalin-fixed patient samples and mouse tissues were processed in an automatic tissue processor, embedded in paraffin (Peloris, Leica) and cut into 4-μm sections. IHC staining was done in an automated staining system (Leica Bond Max, Leica Microsystems, Vista, CA, USA). In brief, slides were deparaffinized and hydrated, and antigen was retrieved by incubating in citrate buffer, pH 6.0, for 1 h with BMP7 (Abcam, Catalog #ab56023), MAPK14 (Thermo Fisher Scientific, Catalog #PA5-17713), SMAD1(Thermo Fisher Scientific, Catalog #38-5400), anti-Phospho-SMAD1/SMAD5 (Ser463, Ser465) (Thermo Scientific–Life Technologies, Catalog #MA5-15124), anti-mannose receptor (CD206) (Abcam, Catalog #ab64693), or anti-CD4 (Bioss, Catalog #bs-0647R) according to the manufacturer's protocol. Slides were examined with a Leica DMI6000B microscope

(Leica, Buffalo Grove, IL), and images were captured by a charge-coupled device camera and imported into the Advanced Spot Image analysis software package and were quantified using Fiji software (http://fiji.sc).

**Reverse phase protein array.** Reverse phase protein array (RPPA) analyses were done by the RPPA-Functional Proteomics core facility at MD Anderson Cancer Center[68]. Tissues were homogenized with a sonicator in a solution containing complete protease and PhosSTOP phosphatase inhibitor cocktail tablets (Roche Applied Science, Mannheim, Germany), 1 mᴍ Na₃VO₄ and lysis buffer (1% Triton X-100, 50 mᴍ HEPES [pH 7.4], 150 mᴍ NaCl, 1.5 mᴍ MgCl₂, 1 mᴍ EGTA, 100 mᴍ NaF, 10 mᴍ NaPPi, 10% glycerol, 1 mᴍ phenylmethylsulfonyl fluoride (serine protease inhibitor), and 10 μg/mL aprotinin). Samples were vortexed frequently on ice and then centrifuged. Cleared supernatants were collected and proteins quantified with a BCA kit (Pierce Biotechnology, Inc., Rockford, IL). Tumor lysates were serially diluted twofold for five dilutions (from undiluted to 1:16 dilution) and arrayed on nitrocellulose-coated slides (Grace Biolab) by Aushon 2470 Arrayer (Aushon BioSystems). Serial diluted lysates were arrayed A total of 5,808 array spots were arranged on each slide, including the spots corresponding to positive and negative controls prepared from mixed cell lysates or dilution buffer, respectively. Antibodies with a Pearson correlation coefficient, between RPPA and western blotting, of >0.7 were considered as "validated". Each slide was probed with a validated primary antibody plus a biotin-conjugated secondary antibody. Multiple replicates of "Control Lysates" on each slide served as a standard for "spatial correction" and "quality test". The QC score from "quality test" indicates good (above 0.8) or poor (below 0.8) antibody staining. Samples were probed with 243 antibodies with a tyramide-based signal amplification approach and visualized by 3, 3′-diaminobenzidine (DAB) colorimetric reaction. The signal obtained was amplified using a Dako Cytomation–catalyzed system (Dako) and visualized by DAB colorimetric reaction. The slides were scanned, analyzed, and quantified using customized software (ArrayPro) to generate spot intensity. The analysis was done in R (version 3.5.1). Normalized data were first log2-transformed (log2($x + 1$)). Proteins expressed at different levels between groups were identified by a $P$ value (obtained from the moderated $t$ statistic from the LIMMA package) of <0.05. To support visual data exploration, a heatmap for the most significant cases was generated by using the heatmap.2 function from the gplots package.

**Protein extraction and western blot analysis.** Total protein was extracted by using NP40 lysis buffer (0.5% NP40, 250 mmol/l NaCl, 50 mmol/l HEPES, 5 mmol/l ethylenediaminetetraacetic acid, and 0.5 mmol/l egtazic acid) supplemented with protease inhibitors cocktails (Sigma-Aldrich). Lysates were centrifuged at $10,000 \times g$

for 10 min, and the supernatant was collected for experiments. Protein lysates (40 μg) were resolved on denaturing gels with 4–20% sodium dodecyl sulfate-polyacrylamide and transferred to nitrocellulose membranes (BioRad Laboratories, Hercules, CA). Membranes were probed with primary antibodies directed against BMP7 (Santa Cruz Biotechnology, Catalog # sc-53917), Phospho-Smad1 (Ser463/465)/ Smad5 (Ser463/465)/ Smad9 (Ser465/467) (Cell Signaling Technologies, Catalog#13820), p38 MAPK (Cell Signaling Technologies, Catalog #8690), Vinculin (Cell Signaling Technologies, Catalog #13901), β-Actin (Cell Signaling Technologies, Catalog # 3700) (dilution 1:500), and a secondary antibody conjugated with horseradish per-oxidase (dilution 1:2000) (Amersham GE Healthcare). The secondary antibody was visualized by using a chemiluminescent reagent (Pierce ECL kit, Thermo Fisher Scientific, Waltham, MA, USA). Uncropped images of blots are shown in Supplementary Fig. 11.

**Isolation of TILs**. Freshly isolated primary tumor tissues (from two or three mice/ group) were washed with ice-cold phosphate-buffered saline (PBS) and digested with 250 μg/mL of Liberase TR (Roche) and 20 μg/mL DNase I (Roche) and incubated for 45 min at 37 °C with shaking. Fetal bovine serum was added, and samples were filtered followed by Histopaque-1077 (Sigma-Aldrich) gradient isolation of TILs. The TILs in the interphase were collected and washed with PBS plus 2% fetal bovine serum. TILs were used for nanostring, quantitative PCR, or flow cytometry analysis.

**NanoString immune profiling**. RNA samples from TILs isolated from 344SQP ($n = 2$ biologically independent samples) or 344SQR ($n = 3$ biologically independent samples) tumors treated with anti-PD1 as previously described were isolated with Triazol (Life Technologies) according to the manufacturer's protocol. Samples with 100 ng of RNA were submitted for NanoString analysis using the PanCancer immune profiling panel of 770 genes at the Genomic and RNA Profiling Core at Baylor College of Medicine. The analysis was done in R (version 3.5.1). The Reporter Code Count data received from the core were preprocessed with the NanoStringNorm package. Genes expressed at different levels between groups were identified by a P value, obtained from the moderated t statistic from the LIMMA package, of <0.05. To support visual data exploration, a heatmap for the most significant cases was generated with the heatmap.2 function from the gplots package.

**Immunofluorescence analysis**. RAW 264.7 cells were counted with a hemocyt-ometer (0.4% Trypan blue solution), diluted to 200,000, and seeded in four-well culture slides (Lab-Tek, Catalog #154917), and allowed to attach overnight. Cells were treated with 250 ng BMP7 (R&D Systems, Catalog #5666-BP-010) or follistatin (R&D Systems, Catalog #769-FS-025) and incubated for 24 h, and then fixed with 1% paraformaldehyde for 10 min, followed by a 10-minute wash in 70% ethanol at room temperature. Cells were then treated with 0.1% NP40 in PBS for 20 min, washed in PBS four times, and then blocked with 5% bovine serum albumin in PBS for 30 min. Cells were then incubated with MAPK14 (L53F8) (Cell Signaling, Catalog #9228) and Phospho-Smad1 (Ser463/465), Smad5 (Ser463/465), and Smad9 (Ser465/467) (D5B10) (Cell Signaling, Catalog #13820) in 5% bovine serum albumin in PBS overnight according to the manufacturer's protocol (dilution 1:25). The next day, cells were incubated with anti-rabbit IgG (H + L), F(ab')2 Fragment (Alexa Fluor 488 Conjugate) (Cell Signaling, Catalog#4412), or Anti-mouse IgG (H + L), F(ab')2 Fragment (Alexa Fluor 488 Conjugate) (Cell Signaling, Catalog #4408) secondary antibody according to the manufacturer's protocol (dilution 1:2000). Then, cells were incubated in the dark with 4,6-diamidino-2-phenylindole dihydrochloride (1 mg/mL) in PBS for 5 min, and coverslips were mounted on a slide with an antifade solution (Molecular Probes; Invitrogen, Waltham, MA). Slides were examined with a fluorescence microscope (Leica, Buffalo Grove, IL), and images were captured by a charge-coupled device camera and imported into the Advanced Spot Image analysis software package.

**Co-culture experiments and treatments**. Viable cells were counted with a hemocytometer (0.4% Trypan blue solution) and diluted to 40,000 cells per well in 24-wells plates. 344SQP, 344SQR, 344SQ ctrl, or 344SQ-shBMP7 cells were seeded at the top inserts (24-mm Transwell with 0.4-μm pore polycarbonate membrane insert, Sigma-Aldrich), and RAW 264.7, peritoneal macrophages or CD4+ T cells were seeded at the bottom of the transwell system. CD4+ T cells were isolated from splenocytes by using Dynabeads Untouched Mouse CD4 Cells Kit (Thermo Fisher Scientific–Life Technologies, Catalog #11416D) and activated with LEAF purified anti-mouse CD3ε antibody (5 μg/mL) and LEAF purified anti-mouse CD28 antibody (1 μg/mL) (Biolegend). Cells were then cultured in complete medium (RPMI-1640 supplemented with 100 units/mL penicillin, 100 μg/mL streptomycin, and 10% heat-inactivated fetal bovine serum) and incubated at 37 °C in 5% $CO_2$ for 24 or 48 h, after which cells were treated with 250 ng of BMP7 (R&D Systems, Catalog #5666-BP-010) or follistatin (R&D Systems, Catalog #769-FS-025) for 24 or 48 h. Follistatin was preincubated with BMP7 at 37 °C for 15 min before their addition to cultures. RNA was then isolated from RAW 264.7 or CD4+ T cells and analyzed for MAPK14, IL1A, IL1B, TNF, CCL5, IL2, and IFNG expression with quantitative PCR.

**In vivo studies**. All mouse studies were approved by the Institutional Animal Care and Use Committee (IACUC) of The University of Texas MD Anderson Cancer Center before their initiation; animal care was provided according to IACUC standards, and all mice had been bred and were maintained in our own specific pathogen-free mouse colony. For RRBS, RPPA, and TILs studies, primary tumors were established by subcutaneous injection of 344SQP or 344SQR cells ($0.5 \times 10^6$ in 100 μL of sterile PBS) into the leg of syngeneic 129 Sv/Ev mice (female, 12–16 weeks old). Mice were then given intraperitoneal injections of anti-PD1 or control IgG antibodies (10 mg/kg) (Bio X cell), starting on day 4 after tumor cell inoculation and continuing twice per week for a total of four doses. At 24 h after the last anti-PD1 treatment, tumor tissues were collected for DNA (two mice/group) and protein isolation (two or three mice/group). For TILs isolation, tumor tissues (three mice/group) were collected a week after the last treatment with anti-PD1. For tumor growth and survival studies, primary tumors were established by subcutaneous injection of 344SQR ctrl or 344SQR shBMP7 cells ($0.5 \times 10^6$ in 100 μL of sterile PBS) into the leg of syngeneic 129 Sv/Ev mice (female, 12–16 weeks old). The mice were then given intraperitoneal injections of anti-PD-1, anti-PDL1 (Durvalumab, Pharmacy MD Anderson), anti-CTLA4 or control IgG antibodies (10 mg/kg) (Bio X cell), starting on day 4 after tumor cell inoculation and continuing twice per week for a total of four doses. At last, primary tumors were established by subcutaneous injection of 344SQR cells ($0.5 \times 10^6$ in 100 μL of sterile PBS) into the leg of syngeneic 129 Sv/Ev mice (female, 12–16 weeks old), which were then given intraperitoneal injections of anti-PD1 (10 mg/kg), control IgG antibodies (10 mg/kg), follistatin (R&D Systems, Catalog #769-FS-025) (0.1 mg/kg) or follistatin (0.1 mg/Kg) plus anti-PD1 (10 mg/kg), starting on day 4 after tumor cell inoculation. Anti-PD1 antibody was given twice per week for a total of four doses; follistatin was given four times per week before and after anti-PD1 for a total of eight doses. Tumors were measured with calipers three times per week and recorded as tumor volume (in $mm^3$) = width$^2$ × length/2. Tumor growth curves were compared with multiple t tests. Mouse survival rates were analyzed by using the Kaplan–Meier method and compared with log-rank tests.

**Flow cytometry**. TILs were blocked with anti-CD16/CD32 (1 μL per sample) before being stained for flow cytometry. For flow cytometry purposes, fluorochrome-conjugated anti-CD3 (Cat #100353), anti-CD4 (Cat #100406), anti-CD8 (Cat #100734), anti-CD45 (Cat #103126), anti-CD11b (Cat #101226), anti-CD11c (Cat #117310), anti-F4/80 (Cat #123108), and anti-CD206 (Cat# 141716) antibodies were purchased from BioLegend. Samples were stained following manufacture's protocol (1 μL of each antibody per sample) and analyzed with an LSR II flow cytometer and FlowJo software (version 10). Gating strategies are shown in Supplementary Fig. 12.

**Statistical analysis**. Prism 8.0 software (GraphPad) and Excel (Microsoft 2016) was used for statistical analyses; the methods used are stated in the figure legends. Statistical significance was accepted at $P \le 0.05$. Student's t tests were used to compare differences between individual groups, and tumor growth curves were compared with multiple t tests, with error bars representing the standard deviation.

**Reporting summary**. Further information on research design is available in the Nature Research Reporting Summary linked to this article.

## Data availability

All data generated or analyzed during this study are included in this published article (and its supplementary information files). Unique material requests should be directed to the corresponding author. Requests are reviewed by MD Anderson Cancer Center to verify whether the request is subject to any intellectual property or confidentiality obligations. Any material that can be shared will be released via a Material Transfer Agreement. Full scans of blots are provided in Supplementary Figs. 12–16. GSE50081 data can be accessed at The Gene Expression Omnibus (GEO) database (https://www.ncbi.nlm.nih.gov/geo/). Clinical information for patients with lung adenocarcinoma was retrieved from the article "An Integrated TCGA Pan-Cancer Clinical Data Resource to Drive High-Quality Survival Outcome Analytics", Cell. Volume 173. (https://www.sciencedirect.com/science/article/pii/S0092867418302290?via%3Dihub), but smoking status. The information regarding smoking status of these was retrieved from cBioPortal for Cancer Genomics (http://www.cbioportal.org/)(Ref: Cerami E et al, The cBio Cancer Genomics Portal: An Open Platform for Exploring Multidimensional Cancer Genomics Data, Cancer Discov. 2012 May;2(5):401-4. https://doi.org/10.1158/2159-8290.CD-12-0095). Gene expression for BMP7 was downloaded as fragments per kilobase millions (FPKM) quantification mRNA-seq data from the Genomic Data Commons Data Portal (https://portal.gdc.cancer.gov/). The Reduced representation bisulfite sequencing (RRBS) data have been deposited in the GEO database and is available at under the GEO accession number GSE154993.

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

## Acknowledgements

We thank Christine F. Wogan, MS, ELS, and Joe D. Dunn of MD Anderson's Division of Radiation Oncology, for editorial contributions. WE thank David Aten, Senior Medical Illustrator of MD Anderson Strategic—Creative Communication for illustration contributions. This work was supported by American Lung Association Award #RG-514395; the Mabuchi Research Fund; the family of M. Adnan Hamed; the Susan and Peter Goodwin Foundation; the Orr Family Foundation (to MD Anderson Cancer Center's Thoracic Radiation Oncology program); and the Wiegand Foundation and the Cancer Center Support (Core) grant CA016672 to The University of Texas MD Anderson Cancer Center from the National Cancer Center, National Institutes of Health.

## Author contributions

M.A.C. and J.W. designed the study, interpreted the data, and wrote the manuscript. F.M. gave technical assistance with the molecular biology and functional analyses in vitro. F.M. and M.A.C. conducted most the experiments. X.W. developed the anti-PD1-resistant mouse model. A.Y., M.S.C., J.E.S., M.D.W., and H.B. assisted with the mouse model studies and immune phenotyping. J.Z. conducted immunohistochemical and pathological analysis. M.R.E. performed the pyrosequencing methylation assays. H.M., R.R., and M.A.C. provided patient data and assisted in statistical analysis. P.H., G.C., and F.S. helped with the design of experiments, coordinated the study and contributed key reagents. Y.L. performed statistical analysis for the reduced representation bilsulfite sequencing results, and C.I. performed statistical analysis for the Nanostring, reverse phase protein array, and The Cancer Genome Atlas results. All authors edited and approved the manuscript.

## Competing interests

J.W.W. reports research support from GlaxoSmithKline, Bristol Meyers Squibb, Merck, Nanobiotix, Mavu Pharma, Takeda, Varian, and Checkmate Pharmaceuticals. J.W.W. serves on the scientific advisory board for Legion Healthcare Partners, RefleXion Medical, MolecularMatch, Merck, AstraZeneca, Aileron Therapeutics, Ventana, OncoResponse, CheckMate Pharmaceuticals, Mavu Pharma, Alpine Immune Sciences, and Nanorobotix. He is co-founder of Healios, MolecularMatch, and OncoResponse and serves as an advisor to Astra Zeneca, Merck, MolecularMatch, Incyte, Aileron, and Nanobiotix. J.W.W. holds stock or ownership in Alpine Immune Sciences, Checkmate Pharmaceuticals, Healios, Mavu Pharma, Legion Healthcare Partners, MolecularMatch, Nanorobotix, OncoResponse, and RefleXion. J.W.W. has accepted honoraria in the form of travel costs from Nanobiotix, RefleXion, Varian, Shandong University, The Korea Society of Radiology, Aileron Therapeutics and Ventana. J.W.W. has the following patents; MP470 (amuvatinib), MRX34 regulation of PDL1, XRT technique to overcome immune resistance. MD Anderson Cancer Center has a trademark for RadScopalTM. P.H. receives laboratory research support from Dragonfly, Immatics, Sanofi, and GlaxoSmithKline. All other authors declare no conflicts of interest.
