## [Peer Review File · Nature Communications]

Reviewers' comments:

Reviewer #1 (Remarks to the Author); expert in BMP

Bone morphogenetic protein 7 promotes resistance to immunotherapy

The authors show that in a mouse tumor cell line (344SQR) that is resistant to anti-PD1 therapy that BMP7 suppresses p38 activity which suppresses inflammatory cytokine expression thereby decreasing the effectiveness of PD1 targeted therapy. The studies were well done and convincing that BMP7 overexpression in this model suppressed the immune response by inhibition of p38. I am not as convinced that blocking BMP7 specifically will demonstrate the same effect in a large number of patient derived tumors. There are many BMP ligands expressed in human tumors such as BMP2 which is the predominant ligand expressed in lung carcinomas. BMP2 has been shown to activate p38 therefore results may change based on ligand expression of the tumor. Even BMP7 at lower concentrations can activate p38. The author suggest because Smad-1/5 is activated that the mechanism of p38 expression is a Smad-1/5 mediated event. This may be true but it is also possible that the p38 regulation is mediated by a Smad-1/5 independent mechanism. TAK1 which is activated by a BMP Smad-1/5 independent mechanism has been shown to activate p38. Although performing a large number of studies on human derived tumors may be beyond the scope of this initial study it should be clearly stated further work is needed to draw firm conclusions. One suggestion would be to perform a set of experiments using a specific BMP receptor inhibitor such as DMH1 in vitro using you mouse model. DMH1 inhibits BMP type I receptors which will block BMP Smad-1/5 dependent signaling. If some of results can be reproduced with a BMP inhibitor it would give greater confidence that targeted BMP therapy could be a means to circumvent PD1 resistance.

In the set of experiments shown in figure 1 authors demonstrate an increase in BMP7 in the PDL1 resistant cell line 344SQR compared to the parental PDL1 sensitive cell line 344SQP. They show 3 tumors of different cell types that progressed with anti-PD1 targeted antibody (Pembrolizumab) or an activating CTLA-4 targeted therapy (Ipilimumab), which both demonstrated greater expression of BMP7. Presented is data that shows the 344SQR resistant cell line is hypomethylated compared to 344SQP cells suggesting that a decrease in methylation is a potential mechanism leading to increased BMP7 expression.

The conclusion drawn that BMP7 creates resistance to anti-PD1 therapy is predominantly based on one mouse cell line. Only one example of a patient derived tumor was actually treated with PD1 targeted therapy and the other two examples were from patients treated with Ipilimumab that activates CTLA-4. Authors do not show BMP7 expression in tumors that responded PD1 targeted therapy. So whether BMP7 creates resistance specifically to anti-PD1 therapy remains a question from these studies. Studies and conclusions from this manuscript were based primarily on the mouse 344SQR and 344SQP cell lines. Human non-small cell lung carcinomas express several BMP ligands and some studies have shown that BMP2 is the most abundant BMP ligand expressed. BMP2 has been shown to activate p38 activity which is opposite to what is shown here.

Data from the series of experiments shown in figure 2 show that BMP7 expression correlates inversely to the expression of p38 in 344SQR and 344SQP cell lines and metastatic tumors that progressed on treatment. Knockdown of BMP7 in 344SQR cell line demonstrated an increase in p38 expression in tumors. Metastatic tumors that progressed on Ipilimumab or Pembrolizumab had a decrease expression of p38 compare to pre-treatment samples. Tumor infiltrating lymphocytes also had a decrease in p38 expression in 344SQR tumors with downregulation of several inflammatory cytokines with a corresponding change in serum cytokines in mice. Treatment of macrophage cell line (RAW 264) demonstrated more p38 expression after BMP7 + follistatin VS BMP7 alone.

Studies presented in figure 3 demonstrate that BMP7 knockdown in 344SQR tumors increased p38 and inflammatory cytokines in TIL. Silencing of p38 in RAW 264.7 cells decreased inflammatory

chemokines. Co-culture experiments with RAW cells showed that knockdown of BMP7 in 344SQR increased inflammatory cytokine production. Follistatin, which was used as a BMP7 inhibitor, also caused an increase in p38 and inflammatory cytokines and chemokines in RAW 264.7 and peritoneal macrophages.

Studies demonstrated in figure 4 demonstrated similar findings with that of mouse derived CD4 cells that BMP7 suppressed p38 and inflammatory cytokine production that was rescued with BMP7 knockdown and follistatin.

In figure 5 studies show tumor xenograft studies in 344SQR treated with follistatin and BMP7 knockdown showed decreased tumor size and improved survival. BMP7 knockdown and follistatin tumors also had increased activated CD4 T (and cell number) cells and decrease in M2 macrophages. All these studies came together nicely making a strong argument that BMP7 is immunosuppressive in their model.

John Langenfeld

Reviewer #2 (Remarks to the Author); expert in mouse models of NSCLC:

In this manuscript, the authors described that BMP7 is responsible for the resistance to anti-PD1 treatment using a murine lung cancer model. Furthermore, they described BMP7 secreted from tumor cells can impact on macrophage and CD4+ T cells through suppressing p38alpha. The link between BMP7 and p38alpha signaling has been reported previously by several reports. In addition, since blocking BMP7 impacts both cancer and immune cells, the contribution from tumor microenvironment versus tumor cells themselves are not clear from the studies presented. It is important to segregate effect from macrophages, CD4+/CD8+ lymphocytes and tumor cells themselves to determine causative effects. There are some additional questions as listed below that need to be addressed to further strengthen the manuscript :

1. Figure 1, the authors showed a panel of differentially regulated genes including BMP7 through comparing PD1 resistant model 344SQR versus control 344SQP model, how about genes other than BMP7 that affected? Are they also involved in drug resistance to anti-PD1? What is their relationship with BMP7? The authors need to discuss this with overall change in the resistant tumors.
2. What is the tumor mutational burden (TMB) status of the model used? Did the authors see differences between high TMB and low TMB tumors for the BMP7 levels in both human patient specimen and mouse model used?
3. Figure 2, as stated above, suppression of p38alpha by BMP7 has been reported previously, thus the presented result is expected. For fig. 2f, the authors described in the manuscript as tumor-infiltrating lymphocytes (TILs). While in the method section, it is described as tumor infiltrating immune cells. Did the authors characterize the infiltrating lymphocytes or total immune cells? In addition, since the method used did not provide detailed information of the composition of extracted cells, it will be helpful to characterize what is composed in the immune infiltrates from both SQR and SQP models.
4. Figure 3, the authors showed a series of fold changes of the cytokine/chemokines examined. It is important to show whether these chemokine/cytokines have the expression levels at physiological meaningful range. Thus the actual concentration of IL-1, TNF-alpha, RANTES showed in fig. 3 need to be presented.
5. Fig. 3g, 3h, the authors showed the co-culturing of tumor cells with macrophages for the cytokine/chemokines examined. Is this contact dependent? Will the tumor supernatant collected from SQR cells after drug treatment have similar effect as co-culturing experiment? Or this effect also requires macrophages with suppressed BMP7 signaling in addition to tumor cells? Again, the actual cytokine/chemokine expression levels need to be examined rather than fold change.
6. Figure 5, the authors showed blocking BMP7 can re-sensitize SQR model response to anti-PD1,

is this effect T cell dependent or macrophage dependent? Fig. 5e only showed total CD4+ T cells, how about changes in Tregs? It is possible that effect observed in T cells is secondary to suppressed Treg in response to the drug treatment.

7. In line with point 6, what is the effect of BMP7 on Treg versus CD4+Foxp3- conventional T cells? Considering BMP7 belong to TGFb family, the effect of BMP7 on Treg will be extremely interesting.

8. Fig 5c and 5e, the IFNgamma+ T cell percentage is quite low in both CD4+ and CD8+ cells, the actual flow panel need to be presented for the robustness of the data.

Reviewer #3 (Remarks to the Author); expert in immunotherapy:

The current manuscript provides a potentially novel and interesting observation that BMP7 can limit anti-tumor immunity through suppression of p38alpha expression in immune effector cells. Although this is an interesting observation, the current data do not fully support the conclusions of the authors and there are several issues that should be addressed.

Major Comments/ Questions

- Throughout the study a number of important experimental controls are omitted which makes it difficult to generate firm conclusions. This is my main criticism of the manuscript.
 - o E.g. Figures 2, B and 2C, 3A, 4B should all have treatment arms with a control antibody for anti-PD-1 (i.e. non-treated). Without these controls it is impossible to ascribe effects to differential responses to anti-PD-1 since the baseline information from non-treated condition is lacking for each respective treatment group.
 - o Similarly Figures 2G, 2H, 3E-3F, 4A and 4F all investigate the role of BMP7 activation in the presence or absence of the BMP7 inhibitor follistatin. However controls from the non-stimulated condition (and follistatin alone) are not shown- so the effects of BMP7 are impossible to deduce in these experimental systems.
 - o Figure 3D the expression of these genes in RAW cells alone is not shown.
- More details on the main experimental system should be provided. The cell lines 344SQP and 344SQR should be introduced more fully in terms of their origin and development. In addition, although they are described as anti-PD-1 responsive and resistant this data is not shown in the manuscript which would be useful to provide context into the magnitude of the phenotype. How resistant are they to anti-PD-1? Do they grow faster in the absence of treatment?
- It would also be nice to support this data using at least one other tumor model to confirm these effects are not specific to this cell line.
- Figures 2C-2E show changes in p38a and pSMAD 1/5/9 expression in PD-1 resistant tumors, BMP7 knockdown tumors and in patients treated with immunotherapy. In all cases conclusions are drawn based on one representative image. Quantification of these effects in multiple mice/ patients are needed to support these claims.
- The experiments with recombinant BMP7 used 250 ng. Is this physiological given the data in Figure 1 indicates the local concentration in serum is <100pg/ml? What level is observed within tumors?
- The correlations shown in Supp Figure 1 are not particularly convincing and it is not clear what the message is supposed to be. In any case, this data may be better used in support of the TILs data presented in Figure 5. Was there any association with CD8/ CD3?
- BMP7 is a known antagonist of TGFbeta (e.g. Zeisberg et al. 2003 PMID 12808448) and since TGFbeta is associated with an immune exclusion phenotype (Mariathasan et al. 2018 PMID 29443960) it would seem important to investigate the role of TGFβ in this system. Could BMP7 expression actually improve outcomes in TGFbeta high/ immune excluded tumours? Does targeting BMP7 have possibly deleterious effects by promoting TGFbeta activity?
- The data in Figure 5B show that Follistatin enhances the efficacy of anti-PD-1. However due to

the above point it seems likely that TGFbeta inhibition could contribute to this therapeutic effect. It would be important to investigate whether efficacy is observed against BMP7 knockdown tumors to determine whether follistatin is operating through this pathway or through some other mechanism e.g. TGFbeta inhibition.

- What is the prognostic significance of BMP7 expression in patient datasets e.g. TCGA?

Minor Comments

- Typing error in line 27. This sentence is currently incomplete, should read 'anti-PD-1 resistant variant'

- Several of the Supplementary Figures show incomplete datasets when more information could easily be provided. For example Supp Figure 3 could show all changes that were observed in comparing PD-1 resistant and responsive tumor models rather than just the magnitude of change for BMP7 which, without context of other genes, loses relevance

- More information should be provided in the results text for the cancer type etc of the patients being treated with immunotherapy e.g. Figure 1G-J.

- Line 123, I believe this sentence should reference Figure 2F not 2D.

- Line 140- It is claimed that the data suggests that "BMP7 regulates p38alpha expression via SMA1 signaling not only in tumors resistant to anti-PD-1 but also in TILs isolated from these tumors relative to control." Is this true? The data to this point (up to Figure 3) only investigated tumors and RAW macrophage cell line so this sentence is not supported.

- Figure legends should provide more details on the experimental setup such as concentrations of reagents etc.

- Figure 4C only warrants a Supplementary Figure as the requirement for p38alpha in cytokine production by T cells is well established.

- Figure 5I. Single agent treatment with anti-CTLA-4 is missing/ not visible.

- More information on the stimulation conditions for the RAW cells needs to be provided in these assays.

Rebuttal letter

We thank the Editor and Reviewers for all their suggestions. We have addressed all concerns as noted below, with Reviewer comments in *black italics* and our replies in red. We believe that our manuscript was immensely improved by all critiques and suggestions, and we hope that our modifications have rendered this work suitable for publication. Thank you.

Reviewer Comments

Reviewer 1 (Remarks to the Author); expert in BMP

Bone morphogenetic protein 7 promotes resistance to immunotherapy

The authors show that in a mouse tumor cell line (344SQR) that is resistant to anti-PD1 therapy that BMP7 suppresses p38 activity which suppresses inflammatory cytokine expression thereby decreasing the effectiveness of PD1 targeted therapy. The studies were well done and convincing that BMP7 overexpression in this model suppressed the immune response by inhibition of p38.

Concern#1: I am not as convinced that blocking BMP7 specifically will demonstrate the same effect in a large number of patient derived tumors.

Reply: The response to BMP7 inhibition in combination with immunotherapy will depend on whether patient tumors overexpress BMP7 and whether BMP7 overexpression is leading to treatment resistance in those patients. The only way to answer this question is to test BMP7 inhibition in tumor samples from patients in clinical trials, and this is what we want to pursue in the future. We were able to test this hypothesis in preclinical models, in which we observed that specific BMP7 knockdown sensitized resistant tumors to anti-PD1, anti-CTLA4, and anti-PDL1 in lung and breast cancer models. Although this is far from the scope of the current study, we believe that BMP7 inhibition in patients with high expression of this protein may sensitize treatment-resistant patients to immunotherapy.

Concern#2: There are many BMP ligands expressed in human tumors such as BMP2 which is the predominant ligand expressed in lung carcinomas. BMP2 has been shown to activate p38 therefore results may change based on ligand expression of the tumor. Even BMP7 at lower concentrations can activate p38.

Reply: We have looked at the expression of BMPs in our global gene profiling data and as shown in **Table 1** below. Although BMP2 may be the predominant ligand expressed in lung carcinomas, our data suggest that BMP7 is the BMP that is upregulated in the setting of resistance to immunotherapies in our model. BMP7 is the most overexpressed BMPs (fold change=8.5) compared with BMP2 (fold change=1.5) and other BMPs, such as BMP2K (fold change=1.3) and BMP5 (fold change=1.25). In addition, induction of BMP7 expression is even higher when tumors are treated with anti-PD1 (fold change=11), and no induction of other BMP is observed as shown in **Table 2** below. We agree that p38 activation may change depending on which BMP is upregulated in a particular tumor. Because BMP7 is overexpressed in our model, our results are in agreement with other studies demonstrating that BMP7 inhibits p38 activation at higher concentrations (PMID:25200314; PMID:18780779).

Table 1. BMPs expression in 334SQR versus 344SQP tumors treated with IgG

Gene.Symbol	Gene.Title	P Value	FDR	FCH
Bmp7	bone morphogenetic protein 7	0.0000110	0.002711	8.514961
Bmp2	bone morphogenetic protein 2	0.000116521	0.007529	1.519223
Bmp2k	BMP2 inducible kinase	0.004547914	0.043688	1.362888
Bmp5	bone morphogenetic protein 5	0.021915008	0.106634	1.254112

Table 2. BMPs expression in 334SQR versus 344SQP tumors treated with anti-PD1

Gene.Symbol	Gene.Title	P Value	FDR	FCH
Bmp7	bone morphogenetic protein 7	0.0000000266	0.000122	11.10651

Concern#3: The author suggest because Smad-1/5 is activated that the mechanism of p38 expression is a Smad-1/5 mediated event. This may be true but it is also possible that the p38 regulation is mediated by a Smad-1/5 independent mechanism. TAK1 which is activated by a BMP Smad-1/5 independent mechanism has been shown to activate p38.

Reply: Great point. Since TAK1 activates p38 via its phosphorylation, we looked at our RPPA data to check p38 α phosphorylation status. As shown below, we did not find changes in p38 α activation between 344SQP versus 344SQR tumors treated with anti-PD1.

Name	344SQR+anti-PD1	344SQP+anti-PD1	FCH	P.Value	FDR
MAPK14(p38_pT180_Y182)	0.85	0.99	-1.10	0.08	0.44

Because we found significant changes in p38 α expression, we then analyzed TAK1 expression in RAW 264.7 cells treated with BMP7 for 24, 48 and 72h. We found no differences on TAK1 expression after treatment with BMP7 as shown below.

Concern#4: Although performing a large number of studies on human derived tumors may be beyond the scope of this initial study it should be clearly stated further work is needed to draw firm conclusions.

Reply: We agree completely, and we have added a sentence on page 16 (Discussion section) to clearly state that more studies are needed to draw firm conclusions.

Concern#5: One suggestion would be to perform a set of experiments using a specific BMP receptor inhibitor such as DMH1 in vitro using you mouse model. DMH1 inhibits BMP type I receptors which will block BMP Smad-1/5 dependent signaling. If some of results can be reproduced with a BMP inhibitor it would give greater confidence that targeted BMP therapy could be a means to circumvent PD1 resistance.

Reply: We agree, and thank you for this suggestion. We tested the BMP receptor inhibitor K02288 in RAW 264.7 cells and in peritoneal macrophages, because this inhibitor blocks ALK2, ALK3, and ALK6, which also bind to BMP7. We first co-cultured macrophages with 344SQR (resistant) cells that secrete high amounts of BMP7 (as shown in **Figure 3F**) and treated them with K02288. We found that, as was true for the follistatin data, this BMP receptor inhibitor promoted expression of Th1 response cytokines. We have added these new data as **Supplementary Figure 5**.

Concern#6: The conclusion drawn that BMP7 creates resistance to anti-PD1 therapy is predominantly based on one mouse cell line.

Reply: We agree. To test whether BMP7 overexpression leads to resistance to immunotherapies in other models, we evaluated whether blocking BMP7 could overcome resistance to anti-PD1 therapy in another mouse model. We chose the 4T1 triple-negative breast cancer mouse model, which is known to be resistant to anti-PD1 antibody treatment. We first established BMP7 knockdown and control 4T1 cells and injected those cells into BALBC syngeneic mice. Tumors were then treated with anti-PD1 or IgG control twice a week for 3 weeks. Our results confirmed that BMP7 knockdown can sensitize breast cancer tumors to anti-PD1 therapy. We added these new data as **Supplementary Figure 7**.

Concern#7: Only one example of a patient derived tumor was actually treated with PD1 targeted therapy and the other two examples were from patients treated with Ipilimumab that activates CTLA-4. Authors do not show BMP7 expression

in tumors that responded PD1 targeted therapy. So whether BMP7 creates resistance specifically to anti-PD1 therapy remains a question from these studies.

Reply: We agree that analyzing BMP7 expression in samples from patients who responded to the immunotherapy would help to validate our mechanism of resistance. Unfortunately our clinical trial protocol allowed us access to biopsy samples only from patients with progression on immunotherapy. We had access to blood samples from patients who responded to immunotherapy and patients who progressed. As shown in **Figure 1f**, patients who progressed on immunotherapy had higher plasma BMP7 levels than did patients with partial response or stable disease. Our preclinical model is also resistant to CTLA4 and PDL1, which gave us the idea to include not only patients treated with pembrolizumab but also those treated with ipilimumab. As shown in **Figure 5h and 5i**, BMP7 knockdown sensitized 344SQR-resistant tumors to both anti-PDL1 and anti-CTLA4 treatment.

Concern#8: *Studies and conclusions from this manuscript were based primarily on the mouse 344SQR and 344SQP cell lines. Human non-small cell lung carcinomas express several BMP ligands and some studies have shown that BMP2 is the most abundant BMP ligand expressed. BMP2 has been shown to activate p38 activity which is opposite to what is shown here.*

Reply: We agree. As discussed in the response to **concern#6**, we evaluated whether blocking BMP7 could overcome resistance to anti-PD1 therapy in the 4T1 triple-negative breast cancer tumor model. Our results confirmed that BMP7 knockdown could sensitize these breast cancer tumors to anti-PD1 therapy. We added these new data as **Supplementary Figure 7**.

As discussed in the response to **concern#2**, BMP7 is highly upregulated in our model relative to other BMPs, including BMP2. Moreover, our data indicate that BMP7 is the only BMP induced after treatment with anti-PD1. Therefore, our findings suggest that specific BMPs can be induced in the setting of cancer immunotherapy resistance. Because BMP7 is overexpressed in our model, our results are in agreement with other studies demonstrating that BMP7 inhibits p38 activation at higher concentrations (PMID:25200314; PMID:18780779).

Reviewer 2 (Remarks to the Author); expert in mouse models of NSCLC:

In this manuscript, the authors described that BMP7 is responsible for the resistance to anti-PD1 treatment using a murine lung cancer model. Furthermore, they described BMP7 secreted from tumor cells can impact on macrophage and CD4+ T cells through suppressing p38alpha.

Concern#1: *The link between BMP7 and p38alpha signaling has been reported previously by several reports.*

Reply: We agree; previous studies have shown that BMP7 can either promote or inhibit p38 MAPK activation (via its phosphorylation) depending on BMP7 concentration (PMID:25200314; PMID:18780779). However, our findings indicate that p38alpha regulation by BMP7 occurs not at the p38 phosphorylation but rather at the transcriptional RNA and protein level. Moreover, our findings showed that BMP7 inhibits p38 expression in macrophages and CD4⁺ T cells to promote resistance to immunotherapy, which has not yet been reported. Our finding that BMP7 downregulates p38 α is new in the context of acquired resistance to cancer immunotherapy.

Concern#2: *In addition, since blocking BMP7 impacts both cancer and immune cells, the contribution from tumor microenvironment versus tumor cells themselves are not clear from the studies presented. It is important to segregate effect from macrophages, CD4+/CD8+ lymphocytes and tumor cells themselves to determine causative effects.*

Reply: Because the combination of BMP7 knockdown with anti-PD1 therapy effectively decreased tumor growth and increased survival in vivo compared with BMP7 knockdown alone (**Figure 5a** and **Supplementary Figure 7**), we focused on the effect of BMP7 secretion by tumor cells on immune cells (CD4⁺, CD8⁺ and macrophages) in the tumor microenvironment. Our findings showed that BMP7 affects activation of both macrophages and CD4⁺ T cells, as demonstrated by a set of experiments involving co-culture with tumor cells and treatment with BMP7 and follistatin (**Figures 3c,d,e,g,h** and **4d,e,f,g**) and immune cells analysis from the tumor microenvironment via IHC and flow cytometry (**Figure 5c-g**). These findings suggest that the mechanism of resistance to anti-PD1 is the secretion of BMP7 in

the tumor microenvironment by tumor cells, which promotes p38 α downregulation in macrophages and CD4⁺ T cells and suppresses tumor immune response.

There are some additional questions as listed below that need to be addressed to further strengthen the manuscript:

Concern#3: 1. Figure 1, the authors showed a panel of differentially regulated genes including BMP7 through comparing PD1 resistant model 344SQR versus control 344SQP model, how about genes other than BMP7 that affected? Are they also involved in drug resistance to anti-PD1? What is their relationship with BMP7? The authors need to discuss this with overall change in the resistant tumors.

Reply: Interesting question. We did not find any papers describing correlation of the significantly hypomethylated or hypermethylated genes with resistance to immunotherapies or with BMP7. The potential role of these genes in resistance to immunotherapies is currently under investigation in our lab. We added a paragraph in the Discussion section page 16 addressing these other genes.

Concern#4: 2. What is the tumor mutational burden (TMB) status of the model used? Did the authors see differences between high TMB and low TMB tumors for the BMP7 levels in both human patient specimen and mouse model used?

Reply: Interesting question. We did not find differences in tumor mutation burden between sensitive and resistant models. We did find transcriptional and epigenetic modifications. Unfortunately the limitations of our clinical protocol precluded us from analyzing TMB analysis in clinical samples.

Concern#5: 3. Figure 2, as stated above, suppression of p38alpha by BMP7 has been reported previously, thus the presented result is expected.

Reply: As discussed in the response to **concern#1**, although previous studies have shown that BMP7 can either activate or inhibit p38 MAPK activation, our study is the first to demonstrate that BMP7 can also regulate p38alpha gene expression at the RNA and protein levels. Our study is also the first to show that BMP7 regulation of p38alpha in immune cells can promote resistance to immunotherapy. These findings were validated in patients with progression on immunotherapies. Therefore, the feature that distinguishes our study from previous ones is our novel data showing that BMP7 downregulates p38alpha expression via SMAD1 pathway in immune cells in the setting of acquired immunotherapy resistance.

Concern#6: For fig. 2f, the authors described in the manuscript as tumor-infiltrating lymphocytes (TILs). While in the method section, it is described as tumor infiltrating immune cells. Did the authors characterize the infiltrating lymphocytes or total immune cells? In addition, since the method used did not provide detailed information of the composition of extracted cells, it will be helpful to characterize what is composed in the immune infiltrates from both SQR and SQP models.

Reply: Thank you for this comment, and the opportunity to correct this error on page 7 and page 28 and Figures legends. We changed the word "lymphocytes" to "leucocytes" to indicate that we included all cells of lymphoid and myeloid lineages.

Details on the composition of TILs (both lymphoid and myeloid populations) in 344SQP and resistant 344SQR tumors were published by our group (*Cancer Research* PMID: 27821490). Because we chose blind revision, the reviewers could not access our published data, and the paper was cited in the manuscript but blinded as "X". We decided to disclose authorship information so the reviewers can access our previous work and check our immune profiling data as requested. We also added a paragraph in the Main section regarding our previous findings on immune cell populations in the 344SQR resistant model (page 3).

Concern#7: 4. Figure 3, the authors showed a series of fold changes of the cytokine/chemokines examined. It is important to show whether these chemokine/cytokines have the expression levels at physiological meaningful range. Thus the actual concentration of IL-1, TNF-alpha, RANTES showed in fig. 3 need to be presented.

Reply: The concentration levels of IL-1 α , IL-1 β , TNF α and RANTES in serum samples from mice bearing 344SQR resistant versus 344SQP sensitive tumors are given in **Supplementary Figure 3**.

Concern#8: 5. Fig. 3g, 3h, the authors showed the co-culturing of tumor cells with macrophages for the cytokine/chemokines examined. Is this contact dependent? Will the tumor supernatant collected from SQR cells after drug treatment have similar effect as co-culturing experiment? Or this effect also requires macrophages with suppressed BMP7 signaling in addition to tumor cells? Again, the actual cytokine/chemokine expression levels need to be examined rather than fold change.

Reply: No, the co-cultures were not contact-dependent. Macrophages were plated at the bottom of 24-well plates and tumor cells were added to the top of the chamber, thereby isolating these cells.

Yes; we found that levels of IL-1 α , IL-1 β , TNF α and RANTES are decreased in serum samples from mice bearing 344SQR resistant versus 344SQP sensitive tumors treated with anti-PD1 (**Supplementary Figure 3**).

Concern#9: 6. Figure 5, the authors showed blocking BMP7 can re-sensitize SQR model response to anti-PD1, is this effect T cell dependent or macrophage dependent?

Reply: Our data showed that BMP7 acts on both macrophages and T cells to suppress anti-tumor immune responses. BMP7 suppressed the expression of Th1 cytokines/chemokines in macrophages and suppressed CD4 activation by decreasing IL-2 and IFN γ expression.

Concern#10: Fig. 5e only showed total CD4+ T cells, how about changes in Tregs? It is possible that effect observed in T cells is secondary to suppressed Treg in response to the drug treatment.

7. In line with point 6, what is the effect of BMP7 on Treg versus CD4+Foxp3- conventional T cells? Considering BMP7 belong to TGF β family, the effect of BMP7 on Treg will be extremely interesting.

Reply: Interesting point. Although some studies have shown that BMPs can promote Tregs, we previously found no differences in Treg populations in 344SQP sensitive tumors relative to 344SQR resistant tumors (PMID: 27821490, Supplementary data).

Concern#11: 8. Fig 5c and 5e, the IFN γ + T cell percentage is quite low in both CD4+ and CD8+ cells, the actual flow panel need to be presented for the robustness of the data.

Reply: We agree; we have supplied the flow data in **Supplementary Data 6**. The percentages were low because we did not stimulate T cells in vitro with PMA/ionomycin and we gated CD8⁺IFN γ ⁺ T cells out of CD45-positive cells. To correct the problem, we reanalyzed the raw data and reported the percentages of CD8⁺IFN γ ⁺ T cells out of CD8⁺ T cells and CD4⁺IFN γ ⁺ T cells out of CD4⁺ T cells for better accuracy (**Figure 5c,e**).

Reviewer 3 (Remarks to the Author); expert in immunotherapy

The current manuscript provides a potentially novel and interesting observation that BMP7 can limit anti-tumor immunity through suppression of p38 α expression in immune effector cells. Although this is an interesting observation, the current data do not fully support the conclusions of the authors and there are several issues that should be addressed.

Major Comments/ Questions

- Throughout the study a number of important experimental controls are omitted which makes it difficult to generate firm conclusions. This is my main criticism of the manuscript.

Reply: We have focused on experiments with anti-PD1 on board since we studied mechanisms of resistance. But we do agree with the reviewer and added controls as explained below.

Concern#1: o E.g. Figures 2, B and 2C, 3A, 4B should all have treatment arms with a control antibody for anti-PD-1 (i.e. non-treated). Without these controls it is impossible to ascribe effects to differential responses to anti-PD-1 since the baseline information from non-treated condition is lacking for each respective treatment group.

Reply: We agree; we have added IgG control data accordingly (**Figures 2b,c, 3a and 4b**).

Concern#2: o Similarly Figures 2G, 2H, 3E-3F, 4A and 4F all investigate the role of BMP7 activation in the presence or absence of the BMP7 inhibitor follistatin. However controls from the non-stimulated condition (and follistatin alone) are not shown- so the effects of BMP7 are impossible to deduce in these experimental systems.

Reply: We agree that adding untreated controls could better show the effect of BMP7 on immune cells in vitro. To address this point, we repeated the experiment in RAW and CD4 T cells and added untreated arm to **Figures 3e and 4f**. We also added new data on RAW and peritoneal macrophages treated with BMP7 compared with untreated control (**Figures 3c,d**). Since we saturated and allowed follistatin to bind and neutralize BMP7 before treating cells, we expected that follistatin alone arm to be similar to untreated control. Therefore, we do not see the need to repeat experiments in **Figure 2h and 4a**.

Concern#3: o Figure 3D the expression of these genes in RAW cells alone is not shown.

Reply: The goal of the experiments depicted in **Figure 3g (former Figure 3d)** was to compare expression of p38 α , IL-1 α/β , TNF α , and RANTES between in RAW cells co-cultured with 344SQP vs 344SQR and 344SQR ctrl and BMP7-knockdown groups.

Concern#4: - More details on the main experimental system should be provided. The cell lines 344SQP and 344SQR should be introduced more fully in terms of their origin and development. In addition, although they are described as anti-PD-1 responsive and resistant this data is not shown in the manuscript which would be useful to provide context into the magnitude of the phenotype. How resistant are they to anti-PD-1? Please refer to our paper PMID:27821490.

Do they grow faster in the absence of treatment? **Reply:** Tumor growth is similar between IgG and anti-PD1 treated tumors, but faster compared with parental tumors, per PMID:27821490.

Information about how anti-PD1-resistant tumors were generated and immune profiling has been published (*Cancer Research* PMID: 27821490). Because we chose blind revision, the reviewers could not access these data. The paper was cited in our manuscript but blinded as "X". Therefore, we decided to disclose authorship information so the reviewers can access our paper and check our immune profiling data as requested. We also added a paragraph to the Main section regarding to our previous findings on immune cell populations in the 344SQR resistant model (page 3).

Concern#5: - It would also be nice to support this data using at least one other tumor model to confirm these effects are not specific to this cell line.

Reply: We agree; to show that BMP7 overexpression leads to resistance to immunotherapies in other models, we evaluated whether blocking BMP7 could overcome resistance to anti-PD1 therapy in another mouse model. We chose the 4T1 triple-negative breast cancer mouse model, which is known to be resistant to anti-PD1 antibody treatment. We first established BMP7 knockdown and control 4T1 cells and injected those cells in BALBC syngeneic mice. Tumors were then treated with anti-PD1 or IgG control twice a week for 3 weeks. Our results confirmed that BMP7 knockdown can sensitize breast cancer tumors to anti-PD1 therapy. We added these new data as **Supplementary Figure 7**.

Concern#6: - Figures 2C-2E show changes in p38 α and pSMAD 1/5/9 expression in PD-1 resistant tumors, BMP7 knockdown tumors and in patients treated with immunotherapy. In all cases conclusions are drawn based on one representative image. Quantification of these effects in multiple mice/ patients are needed to support these claims.

Reply: We agree, and we have added quantification of IHC images to **Supplementary Figures 2**.

Concern#7: - The experiments with recombinant BMP7 used 250 ng. Is this physiological given the data in Figure 1 indicates the local concentration in serum is <100pg/ml? What level is observed within tumors?

Reply: We used a concentration of 250 ng BMP7 for our in vitro studies based on previous studies that tested different concentrations in macrophages in vitro (PMID: 24376781). Serum BMP7 levels from mice and patients and levels in media in vitro are depicted in **Figure 1e,f** and **Figure 3f**.

Concern#8: - The correlations shown in Supp Figure 1 are not particularly convincing and it is not clear what the message is supposed to be. In any case, this data may be better used in support of the TILs data presented in Figure 5. Was there any association with CD8/ CD3?

Reply: The correlations shown in former **Supplementary Figure 1** are from public raw data and can be confirmed by others. Our goal was to evaluate the correlation between BMP7 expression and immune-related markers in samples from patients with non-small cell lung cancer (TCGA-LUAD). We reviewed these data and did a new analysis of another data set as explained below (in response to Concern#10). No correlation was found with CD8/CD3, but we found an inverse correlation between BMP7 and p38 α (MAPK14) and a positive correlation between BMP7 expression and CD68 (resident macrophages marker) and FOXP3 (T regulatory cell marker), as shown below. We moved these new data to **Supplementary Figure 8** to support **Figure 5** as suggested.

Concern#9: - BMP7 is a known antagonist of TGFbeta (e.g. Zeisberg et al. 2003 PMID 12808448) and since TGFbeta is associated with an immune exclusion phenotype (Mariathasan et al. 2018 PMID 29443960) it would seem important to investigate the role of TGFbeta in this system. Could BMP7 expression actually improve outcomes in TGFbeta high/ immune excluded tumours? Does targeting BMP7 have possibly deleterious effects by promoting TGFbeta activity? The data in Figure 5B show that Follistatin enhances the efficacy of anti-PD-1. However due to the above point it seems likely that TGFbeta inhibition could contribute to this therapeutic effect. It would be important to investigate whether efficacy is observed against BMP7 knockdown tumors to determine whether follistatin is operating through this pathway or through some other mechanism e.g. TGFbeta inhibition.

Reply: Thank you for these interesting observations. At your suggestion, we analyzed serum TGFbeta levels from mice bearing parental (344SQP) or resistant (344SQR) tumors treated with IgG control or anti-PD1 antibodies by ELISA. As shown in the [Figure](#) below, we did not find differences in TGFbeta levels in 344SQP vs. 344SQR treated with IgG control (P=0.128) or 344SQP vs. 344SQR treated with anti-PD1 antibodies (P=0.135).

Concern#10: What is the prognostic significance of BMP7 expression in patient datasets e.g. TCGA?

Reply: Thank you for this question, which led us to some interesting findings. In the TCGA lung adenocarcinoma cohort, we found BMP7 to be significant in univariate Cox analysis but not in a multivariate Cox model that included BMP7 and disease stage.

Table 1a. TCGA LUAD cohort Univariate analysis

Variable	HR	lower .95	upper .95	P value	no pt.
Age (continuous)	0.999	0.98	1.01	0.88914	422

Smoking (Ever vs Never)	0.87	0.55	1.37	0.55186	419
Sex (Male vs Female)	0.95	0.69	1.31	0.77607	431
Stage (III-IV vs I-II)	2.64	1.88	3.69	0.00000002	423
BMP7 (FPKM)(continuous)	1.19	1.02	1.39	0.0247	431

Table 1b. TCGA LUAD cohort Multivariate analysis including disease stage & BMP7 (n=423)

Variable	HR	lower .95	upper .95	P value
Stage (III-IV vs I-II)	2.57	1.83	3.6	0.00000005
BMP7 (FPKM)(continuous)	1.14	0.98	1.32	0.09778

Because the *P* value for BMP7 was near-significant in the multivariate model, we searched GEO for a second group consisting of at least 100 patients with lung adenocarcinoma. We found GSE50081 with 181 stage I and II NSCLC cases (from Der SD et al, Validation of a histology-independent prognostic gene signature for early-stage, non-small-cell lung cancer including stage IA patients, *J Thorac Oncol.* 2014 Jan;9(1):59-64. doi: 10.1097/JTO.000000000000042, PMID: 24305008). We retrieved microarray (Affymetrix Human Genome U133 Plus 2.0 Array) expression (normalized log2) data for BMP7 along with clinical information on the patients. Among those 181 cases, 127 were adenocarcinomas. We performed univariate and multivariate Cox analysis and found that BMP7 (21160_at) was an independent marker of poor overall survival.

Table 2a. Der SD et al, LUAD cohort Univariate analysis (n=127)

Variable	HR	lower .95	upper .95	P value
Sex	1.41	0.81	2.46	0.22804
Stage (II vs I)	2.44	1.38	4.32	0.00210
Age (continuous)	1.02	0.99	1.05	0.19195
Smoking (Ever vs Never)	1.68	0.75	3.77	0.20721
BMP7 (209590_at) (continuous)	1.17	0.91	1.52	0.22187
BMP7 (209591_s_at) (continuous)	1.16	1.00	1.35	0.04274
BMP7 (211259_s_at) (continuous)	1.40	0.89	2.20	0.14535
BMP7 (211260_at) (continuous)	5.40	1.70	17.15	0.00426
BMP7 (233583_at) (continuous)	1.12	0.29	4.29	0.86452

Table 2b. Der SD et al, LUAD cohort Multivariate analysis Stage & BMP7 (211260_at) (n=127)

Variable	HR	lower .95	upper .95	P value
Stage (II vs I)	2.42	1.37	4.29	0.00234
BMP7 (211260_at) (continuous)	5.43	1.68	17.53	0.00463

Next, to visualize the survival difference for BMP7 (21160_at), we used the log-rank test to identify a cut-off point for the most significant (lowest *P* value) split in the high- vs low-mRNA level groups. That cut-off was 0.65. Patients at risk in the low- and high-mRNA groups at different times are shown at the bottom of the graph below, and the median survival in each group are shown with these data in **Supplementary Figure 9.**

Minor Comments

- Typing error in line 27. This sentence is currently incomplete, should read 'anti-PD-1 resistant variant'

Reply: Thank you for pointing out this error, which we have corrected per your suggestion.

- Several of the Supplementary Figures show incomplete datasets when more information could easily be provided. For example Supp Figure 3 could show all changes that were observed in comparing PD-1 resistant and responsive tumor models rather than just the magnitude of change for BMP7 which, without context of other genes, loses relevance

Reply: Supplementary Figure 3 depicts multiplex ELISA for cytokines/chemokines in serum samples from mice bearing resistant tumors versus sensitive tumors. Perhaps the reviewer is referencing Supplementary Table 2, which displays BMP7 expression profiling data for sensitive versus resistant models treated with PD1. If that is the case, our focus on validating BMP7 as an immunotherapeutic target for resistance led us to include only its expression in the Supplementary data to validate our methylation, quantitative PCR, and IHC data.

- More information should be provided in the results text for the cancer type etc of the patients being treated with immunotherapy e.g. Figure 1G-J.

Reply: Cancer type, age, sex, disease, treatment, and response are depicted in Supplementary Tables 4 and 5.

- Line 123, I believe this sentence should reference Figure 2F not 2D.

Reply: With our thanks for finding this mistake, we have corrected it accordingly.

- Line 140- It is claimed that the data suggests that "BMP7 regulates p38alpha expression via SMA1 signaling not only in tumors resistant to anti-PD-1 but also in TILs isolated from these tumors relative to control." Is this true? The data to this point (up to Figure 3) only investigated tumors and RAW macrophage cell line so this sentence is not supported.

Reply: We agree, and we corrected the statement in question as follows: "These results suggest that BMP7 regulates p38 α expression via SMAD1 signaling not only in tumors resistant to anti-PD1 but also in immune cells" on page 9.

- Figure legends should provide more details on the experimental setup such as concentrations of reagents etc.

Reply: We followed Nature Communications guidelines for Figure Legends as stated below, with minimum methodological details and under 350 words. Nonetheless, we added concentration of reagents to the legends as suggested.

"Figure legends

Figure legends begin with a brief title sentence for the whole figure and continue with a short description of what is shown in each panel and the symbols used; methodological details should be kept to a minimum as much as possible. Each legend should total no more than 350 words. Text for figure legends should be provided in numerical order after the references."

- Figure 4C only warrants a Supplementary Figure as the requirement for p38alpha in cytokine production by T cells is well established.

Reply: We agree; we have added this data set to **Supplementary Figure 4**.

- Figure 5I. Single agent treatment with anti-CTLA-4 is missing/ not visible.

Reply: With our thanks for pointing out this issue, we have corrected **Figure 5I** accordingly.

- More information on the stimulation conditions for the RAW cells needs to be provided in these assays.

Reply: Information on studies of the RAW cells is described in the Methods on page 30.

Reviewers' comments:

Reviewer #1 (Remarks to the Author):

The authors have answered all my questions and I have no others concerns.

Reviewer #2 (Remarks to the Author):

The authors have addressed most of my comments.

There are only two remaining issues that if addressed will further strengthen the findings of the manuscript.

Original Concern#2: In addition, since blocking BMP7 impacts both cancer and immune cells, the contribution from tumor microenvironment versus tumor cells themselves are not clear from the studies presented. It is important to segregate effect from macrophages, CD4+/CD8+ lymphocytes and tumor cells themselves to determine causative effects.

Reply: Because the combination of BMP7 knockdown with anti-PD1 therapy effectively decreased tumor growth and increased survival in vivo compared with BMP7 knockdown alone (Figure 5a and Supplementary Figure 7), we focused on the effect of BMP7 secretion by tumor cells on immune cells (CD4+, CD8+ and macrophages) in the tumor microenvironment. Our findings showed that BMP7 affects activation of both macrophages and CD4+ T cells, as demonstrated by a set of experiments involving co-culture with tumor cells and treatment with BMP7 and follistatin (Figures 3c,d,e,g,h and 4d,e,f,g) and immune cells analysis from the tumor microenvironment via IHC and flow cytometry (Figure 5c-g). These findings suggest that the mechanism of resistance to anti-PD1 is the secretion of BMP7 in the tumor microenvironment by tumor cells, which promotes p38 α downregulation in macrophages and CD4+ T cells and suppresses tumor immune response.

Reviewer follow-up comment: One the most important findings/claims in this manuscript is that BMP7-knockdown tumors treated with IgG or anti-PD1 had decreased percentages of M2 macrophages compared with control tumors treated with IgG or anti-PD1 by immune profiling (Fig 5d) and by IHC (Fig 5f). However, it is very difficult to make a solid conclusion, as there was no significance shown in Fig 5d and no qualifications shown in Fig 5f. Moreover, across the whole manuscript, for most in vivo studies, such as flow cytometry analysis etc (Fig 3, 4 and 5), the sample size seems to be a bit small (some groups n = 3, while others n =2).

Original Concern#9: In Figure 5, the authors showed blocking BMP7 can re-sensitize SQR model response to anti-PD1, is this effect T cell dependent or macrophage dependent?

Reply: Our data showed that BMP7 acts on both macrophages and T cells to suppress anti-tumor immune responses. BMP7 suppressed the expression of Th1 cytokines/chemokines in macrophages and suppressed CD4 activation by decreasing IL-2 and IFN γ expression.

Reviewer follow-up comment: the authors still did not provide direct/strong in vivo evidence that the observed effect is directly through CD4 T cells or macrophages. A potential experiment to consider could be depleting these responsive immune cells.

Reviewer #3 (Remarks to the Author):

The authors have addressed some of my concerns and in particular the inclusion of the TCGA significantly improves the significance of the study. However some questions still remain.

In terms of the experimental controls referred to in my original review. The inclusion of these additional controls improves the clarity of the manuscript, however in some instances this could be further improved by combining the data onto one graph. For example Figure 3A and 3B include the data from the Ig and anti-PD-1 treated mice on separate graphs. Can these data be combined? The impression of the data presented is that the effect is not related to a response to anti-PD-1 per se but rather an increase in cytokines/ chemokines mediated by BMP 7 regardless of PD-1 treatment. This is an important message to convey to the reader.

Re: Concern 9. The measurement of TGFbeta in the serum (and lack of significant differences) does not exclude the possibility that BMP7 targeting/ follistatin is modulating the activity of TGFbeta, particularly in the tumor microenvironment. This may be beyond the current study but should at least be mentioned in the discussion to place these findings into the context of this pathway.

The increased efficacy observed in the 4T1 model were relatively mild which somewhat limits the significance of the study. Admittedly this model is very difficult to treat with immune checkpoint blockade and so either more aggressive treatment (e.g. Combined anti-PD-1/ anti-CTLA-4) in this model, or the use of a more responsive model may be warranted to confirm the broad utility of targeting this pathway in multiple models.

Minor points

Re: Concern 7. Although the references paper indicates that similar concentrations of BMP7 were used in this study it does not address whether this concentration is likely to occur within the tumor microenvironment. The data referred to by the authors in Figure 1 and Figure 3 are 1000x lower than the concentrations used in vitro. This data does not support the use of such high concentrations unless the authors have data/ previous evidence to suggest that higher concentrations are achieved in the tumor.

Re Concern 8. Apologies if my statement 'not particularly convincing' was misconstrued. I do not doubt the analysis (as the authors outline this is publically available data), but rather the message. I was referring to the extent of correlation between BMP7 expression and markers for macrophages and Tregs which I do not find compelling.

Rebuttal letter

Reviewer #2 (Remarks to the Author):

The authors have addressed most of my comments.

There are only two remaining issues that if addressed will further strengthen the findings of the manuscript.

Original Concern#2: In addition, since blocking BMP7 impacts both cancer and immune cells, the contribution from tumor microenvironment versus tumor cells themselves are not clear from the studies presented. It is important to segregate effect from macrophages, CD4+/CD8+ lymphocytes and tumor cells themselves to determine causative effects.

Reply: Because the combination of BMP7 knockdown with anti-PD1 therapy effectively decreased tumor growth and increased survival in vivo compared with BMP7 knockdown alone (Figure 5a and Supplementary Figure 7), we focused on the effect of BMP7 secretion by tumor cells on immune cells (CD4+, CD8+ and macrophages) in the tumor microenvironment. Our findings showed that BMP7 affects activation of both macrophages and CD4+ T cells, as demonstrated by a set of experiments involving co-culture with tumor cells and treatment with BMP7 and follistatin (Figures 3c,d,e,g,h and 4d,e,f,g) and immune cells analysis from the tumor microenvironment via IHC and flow cytometry (Figure 5c-g). These findings suggest that the mechanism of resistance to anti-PD1 is the secretion of BMP7 in the tumor microenvironment by tumor cells, which promotes p38 α downregulation in macrophages and CD4+ T cells and suppresses tumor immune response.

Reviewer follow-up comment: One the most important findings/claims in this manuscript is that BMP7-knockdown tumors treated with IgG or anti-PD1 had decreased percentages of M2 macrophages compared with control tumors treated with IgG or anti-PD1 by immune profiling (Fig 5d) and by IHC (Fig 5f). However, it is very difficult to make a solid conclusion, as there was no significance shown in Fig 5d and no qualifications shown in Fig 5f.

Replay: Although the p value is not significant on cytometry analysis due to variability among mice, there is a clear trend showing decrease on M2 macrophages in BMP7-knockdown or BMP7-knockdown plus anti-PD1 compared to IgG or PD1 groups. As per suggestion, we quantified IHC using Optical intensity using Fuji software. Our results showed significant decrease on M2 macrophages in BMP7-knockdown or BMP7-knockdown plus anti-PD1 compared to IgG or PD1 groups as shown in Supplemental Figure 8. These results is in accordance with our flow cytometry data.

Moreover, across the whole manuscript, for most in vivo studies, such as flow cytometry analysis etc (Fig 3, 4 and 5), the sample size seems to be a bit small (some groups n = 3, while others n =2).

Replay: We believe that there is a confusion in the statement above. The reviewer is possibly referring to gene expression analysis in TILs which was n=3 mice for IgG group and n=2 mice for PD1 group in Figures 3B and 4C. For all in vivo studies involving tumor growth analysis we used more than 3 mice per group, which were n=5 for studies depicted in Figures 5A, 5B and 5h); n=6 (Supplemental Figure 6) and n=8 (Figure 5i) mice per group as per Figure legends. For the new depletion studies in Supplemental

Figure 9, we used n=4 or n=5 per group. Furthermore, we used n=3 mice per group for Flow cytometer analysis, also depicted in Figure legends.

Original Concern#9: In Figure 5, the authors showed blocking BMP7 can re-sensitize SQR model response to anti-PD1, is this effect T cell dependent or macrophage dependent?

Reply: Our data showed that BMP7 acts on both macrophages and T cells to suppress anti-tumor immune responses. BMP7 suppressed the expression of Th1 cytokines/chemokines in macrophages and suppressed CD4 activation by decreasing IL-2 and IFN γ expression.

Reviewer follow-up comment: the authors still did not provide direct/strong in vivo evidence that the observed effect is directly through CD4 T cells or macrophages. A potential experiment to consider could be depleting these responsive immune cells.

Replay: We agree. To address this question, we performed depletion experiments in vivo as per your suggestion. We used specific antibodies to deplete CD4 or macrophages in BMP7-knockdown tumors treated with anti-PD1. As shown on Supplementary Figure 9, CD4 depletion completely reverted anti-PD1 response seen in BMP7-knockdown treated tumors. There was no difference between tumors treated with anti-PD1 alone or in combination with macrophages depleting antibody. These results might be explained by the fact that M2 macrophages were also depleted which might improve anti-tumor immune response.

Reviewer #3 (Remarks to the Author):

The authors have addressed some of my concerns and in particular the inclusion of the TCGA significantly improves the significance of the study. However some questions still remain.

In terms of the experimental controls referred to in my original review. The inclusion of these additional controls improves the clarity of the manuscript, however in some instances this could be further improved by combining the data onto one graph. For example Figure 3A and 3B include the data from the Ig and anti-PD-1 treated mice on separate graphs. Can these data be combined? The impression of the data presented is that the effect is not related to a response to anti-PD-1 per se but rather an increase in cytokines/ chemokines mediated by BMP 7 regardless of PD-1 treatment. This is an important message to convey to the reader.

Replay: We believe that the data is more accurate if we separate IgG and anti-PD1 groups. BMP7 decrease the expression of p38 and p38-related cytokines/chemokines independently of anti-PD1 treatment as also confirmed in in vitro studies where anti-PD1 is not present.

Re: Concern 9. The measurement of TGFbeta in the serum (and lack of significant differences) does not exclude the possibility that BMP7 targeting/ follistatin is modulating the activity of TGFbeta, particularly in the tumor microenvironment. This may be beyond the current study but should at least be mentioned in the discussion to place these findings into the context of this pathway.

Replay: We agree. We added a sentence on the text as per your suggestion on page 8.

The increased efficacy observed in the 4T1 model were relatively mild which somewhat limits the significance of the study. Admittedly this model is very difficult to treat with immune checkpoint blockade and so either more aggressive treatment (e.g. Combined anti-PD-1/ anti-CTLA-4) in this model, or the use of a more responsive model may be warranted to confirm the broad utility of targeting this pathway in multiple models.

Replay: As mentioned by the reviewer, 4T1 is known to be very resistance to immunotherapy. This is the main reason why we have selected this model to validate our findings. Previous studies have shown that 3 out of 10 mice respond to PD1 and CTLA-4 combination in 4T1 models (PMID: 25071169). In addition, the comparison between PD1 plus CTLA-4 is out of the scope of this study. Since our study is about resistance to immunotherapies, we do not see a clear rationale to use a model that is more responsive to immunotherapies in order to validate our findings. Although BMP7-knockdown plus anti-PD1 therapy do not completely cure mice from tumors in 4T1 model, the tumor size is significantly decreased compared to control (P=0.016). In addition, the combination of BMP7-knockdown plus anti-PD1 therapy clearly improved survival as shown in Supplemental Figure 6 (P=0.001).

Minor points

Re: Concern 7. Although the references paper indicates that similar concentrations of BMP7 were used in this study it does not address whether this concentration is likely to occur within the tumor microenvironment. The data referred to by the authors in Figure 1 and Figure 3 are 1000x lower than the concentrations used in vitro. This data does not support the use of such high concentrations unless the authors have data/ previous evidence to suggest that higher concentrations are achieved in the tumor.

Replay: The concentration showed in Figure 1 is not local, but systemic concentration in blood circulating in the whole body of mice or patients. Therefore, it should be more diluted compared to BMP7 concentration at the tumor site. In addition, the concentration showed in Figure 3 is from media produced by 300,000 cells. A human tumor mass has 100,000 cells in 1 mm (PMID:22514490). Tumors in mice that were collected for analysis were in average size around 1000-2000 mm³. Therefore, it had theoretically twice as much as 100,000,000-200,000,000 cells since human cancer cells are 20um in average compared to mouse cancer cells which are 10um in average (PMID:28060956). Consequently, it should produce 1000x higher concentration of BMP7 compared to cells in vitro. This explains why BMP7 levels in media are lower compared to the concentration we selected for testing in immune cells in vitro. More importantly, we observed the same effect on p38alpha and p38-alpha regulated cytokines/chemokines expression in both immune cells treated with 250 ng in vitro and TILs collected from average 1000-2000 mm³ tumors from in vivo model.

Re Concern 8. Apologies if my statement 'not particularly convincing' was misconstrued. I do not doubt the analysis (as the authors outline this is publically available data), but rather the message. I was referring to the extent of correlation between BMP7 expression and markers for macrophages and Tregs which I do not find compelling.

Replay: We understand the reviewer point of view, but we also believe that is important to validate our findings in patient's data to strengthen our findings. The message is that patients that overexpress BMP7 also present higher infiltration of immunosuppressive cells which supports our findings showing that BMP7 can promote resistance to immunotherapies.

REVIEWERS' COMMENTS:

Reviewer #2 (Remarks to the Author):

the authors have adequately addressed all my comments

Reviewer #3 (Remarks to the Author):

My concerns remain and are:

- 1) The data in Figure 3 suggest that BMP7 inhibits anti-tumor immunity regardless of treatment with anti-PD-1. This is at odds with the title and main message of the manuscript. This data is all shown as relative expression and so it is impossible to evaluate these results between control and anti-PD-1 treated groups. Raw data should at least be shown as supplementary information.
- 2) Significance- The authors propose that targeting this pathway could enhance response to immunotherapy. But the therapeutic data/ models currently presented do not make a compelling case for this.
- 3) Role for BMP7 in suppressing immune cells at physiological concentrations; whilst it is feasible/ likely that BMP7 is more abundant in the tumor relative to blood this needs to be demonstrated, or a clear reason why this cannot be done should be stated.

Response to Reviewer 3

“My concerns remain and are:

“1) The data in Figure 3 suggest that BMP7 inhibits anti-tumor immunity regardless of treatment with anti-PD-1. This is at odds with the title and main message of the manuscript.”

Reply: With our apologies for the lack of clarity, the title and main message of the study are that PD1-resistant tumors overexpress BMP7, which counteracts the effects of PD1 treatment and inhibits antitumor immunity. Our PD1-resistant model was generated from a PD1-sensitive cell line, 344SQP. Tumors were passaged under anti-PD1 antibody pressure, and resistant clones were isolated. During this process of acquiring resistance to anti-PD1, we noted changes that included hypomethylation in the BMP7 promoter (Figure 1) that led to this gene being overexpressed, which in turn promoted immunosuppression in the tumor microenvironment. Notably, we also saw BMP7 overexpression in samples from patients who did not respond to immunotherapies. With that said, after the model was established, BMP7 overexpression occurs constitutively, with no need for anti-PD1 to be present. Thus, it was expected that BMP7 would be overexpressed once resistance was established and would continue to be overexpressed in experiments involving IgG or PD1 treatment.

“This data is all shown as relative expression and so it is impossible to evaluate these results between control and anti-PD-1 treated groups.”

Reply: We undertook the analysis shown in Figure 3 to simply show that BMP7 decreased the expression of genes in inflammatory pathways, with no intent to compare IgG or PD1 groups because BMP7 would be overexpressed in either case. We believe that results from quantitative RT-PCR can be appropriately displayed as mRNA relative expression or fold change.

“Raw data should at least be shown as supplementary information.”

Reply: We added raw data to Figure 3 as requested.

“2) Significance- The authors propose that targeting this pathway could enhance response to immunotherapy. But the therapeutic data/ models currently presented do not make a compelling case for this.”

Reply: We were able to show that BMP7 is overexpressed in a PD1-resistant model of lung adenocarcinoma and in samples from patients who do not respond to immunotherapies. We further found that circulating BMP7 in the blood was also increased in both the model and patients. BMP7 suppresses inflammatory pathways in macrophages by targeting p38 and consequently decrease production of proinflammatory cytokines such as TNF- α , IL-1 β , IL-1 α and RANTES. BMP7 also suppresses activation of CD4+ T cells by downregulating p38 and IL-2 and IFN- γ production. BMP7 knockdown in lung adenocarcinoma re-sensitized resistant tumors to anti-PD1 and anti-CLTA-4. BMP7 knockdown in triple-negative breast cancer also re-sensitized resistant tumors to anti-PD1. More importantly, we were able to show that

BMP7 depends on CD4+ T cells to promote resistance to PD1 inhibitors. We contend that collectively, these findings show that targeting BMP7 in combination with PD1 inhibitors may overcome resistance to immunotherapies.

“3) Role for BMP7 in suppressing immune cells at physiological concentrations; whilst it is feasible/ likely that BMP7 is more abundant in the tumor relative to blood this needs to be demonstrated, or a clear reason why this cannot be done should be stated.”

Reply: In Figures 3A,B and 4B,C, immune cells from tumors (in vivo setting) were isolated in the presence (ctrl) or absence of BMP7 (shBMP7). In those experiments, BMP7 was present at physiological concentrations, which confirmed the feasibility of the assay. The results showed that upon BMP7 knockdown led to upregulation of p38 and p38-mediated pro-inflammatory cytokines. These results were further confirmed by the in vitro studies, where BMP7 was also present at physiological concentrations, in Figures 3G and 4D.